EMBO
Molecular Medicine

# PD-L1 blockade enhances response of pancreatic ductal adenocarcinoma to radiotherapy

Abul Azad[1],[†], Su Yin Lim[1],[†], Zenobia D'Costa[1], Keaton Jones[1], Angela Diana[1], Owen J Sansom[2], Philipp Kruger[3], Stanley Liu[4], W Gillies McKenna[1], Omer Dushek[3], Ruth J Muschel[1] & Emmanouil Fokas[1],[‡],[§],[¶],[*]

## Abstract

Pancreatic ductal adenocarcinoma (PDAC) is considered a non-immunogenic tumor, and immune checkpoint inhibitor monotherapy lacks efficacy in this disease. Radiotherapy (RT) can stimulate the immune system. Here, we show that treatment of KPC and Pan02 murine PDAC cells with RT and gemcitabine upregulated PD-L1 expression in a JAK/Stat1-dependent manner. *In vitro*, PD-L1 inhibition did not alter radio- and chemosensitivity. *In vivo*, addition of anti-PD-L1 to high (12, 5 × 3, 20 Gy) but not low (6, 5 × 2 Gy) RT doses significantly improved tumor response in KPC and Pan02 allografts. Radiosensitization after PD-L1 blockade was associated with reduced CD11b⁺Gr1⁺ myeloid cell infiltration and enhanced CD45⁺CD8⁺ T-cell infiltration with concomitant upregulation of T-cell activation markers including CD69, CD44, and FasL, and increased CD8:Treg ratio. Depletion of CD8⁺ T cells abrogated radiosensitization by anti-PD-L1. Blockade of PD-L1 further augmented the effect of high RT doses (12 Gy) in preventing development of liver metastases. Exploring multiple mathematical models reveals a mechanism able to explain the observed synergy between RT and anti-PD-L1 therapy. Our findings provide a rationale for testing the use of immune checkpoint inhibitors with RT in PDAC.

**Keywords** liver metastases; mathematical modeling; pancreatic cancer; PD-L1 immune checkpoint; radiosensitization
**Subject Categories** Cancer; Digestive System; Immunology

See also: **J Dörrie** (February 2017)

## Introduction

Pancreatic ductal adenocarcinoma (PDAC) is a deadly malignancy with a 5-year survival rate of 5% (Hidalgo, 2010). Surgery is the only curative option, but the majority of patients are diagnosed at an advanced or metastatic stage. Also, PDAC is refractory to radiotherapy (RT) and chemotherapy (Ryan *et al*, 2014). Therefore, new therapeutic strategies are urgently required to improve survival in PDAC.

The immune checkpoint programmed death 1 (PD-1) receptor and its ligand PD-L1 (Iwai *et al*, 2002) are often activated in cancer and play an important role in mediating escape from immune control by inhibiting cytotoxic T-cell function (Tolaney *et al*, 2015). Inhibition of the PD-1/PD-L1 axis has produced impressive response rates in various malignancies, such as melanoma, renal and non-small cell lung cancer (Iwai *et al*, 2002; Topalian *et al*, 2012; Hamid *et al*, 2013). However, although PD-L1 is expressed in human PDAC samples (Spivak-Kroizman *et al*, 2013; Hutcheson *et al*, 2016), monotherapy with PD-L1 inhibitors lacked efficacy in this disease (Brahmer, 2012). This can be attributed to the fact that PDAC is a "non-immunogenic" tumor characterized by low mutational burden (Liu *et al*, 2011), lack of CD8⁺ T-cell infiltration (Ino *et al*, 2013), and the presence of immunosuppressive myeloid cell populations (Dunn *et al*, 2002; Stromnes *et al*, 2015).

Radiotherapy can activate the immune system to trigger an antitumor immune response following cytotoxic death and release of immunostimulating signals that can increase trafficking of T cells to the tumor (Burnette & Weichselbaum, 2013; Formenti & Demaria, 2013). Although studies have proposed the use of immunomodulators to enhance response to RT in various tumor types (Sharabi *et al*, 2015a), the efficacy of this concept in PDAC remains unexplored. Because PDAC is considered non-immunogenic (Hiraoka *et al*, 2006; Lutz *et al*, 2014) and thus less responsive to immunotherapies alone, we hypothesized that RT and anti-PD-L1 might have synergistic roles in the treatment of this disease. Here, we demonstrate that blockade of PD-L1 significantly improves tumor response to high but not low radiation doses in different

1 Department of Oncology, CRUK/MRC Oxford Institute for Radiation Oncology, University of Oxford, Oxford, UK
2 CRUK Beatson Cancer Institute, University of Glasgow, Glasgow, UK
3 Sir William Dunn School of Pathology, University of Oxford, Oxford, UK
4 Department of Radiation Oncology, Sunnybrook Research Institute, Sunnybrook Health Sciences Centre, University of Toronto, Toronto, ON, Canada
*Corresponding author. Tel: +44 1865 225834; Fax: +44 1865 857127; E-mail: emmanouil.fokas@oncology.ox.ac.uk
†These authors contributed equally to this work
‡Present address: Department of Radiotherapy and Oncology, Goethe University Frankfurt, Frankfurt, Germany
§Present address: German Cancer Research Center (DKFZ), Heidelberg, Germany
¶Present address: German Cancer Consortium (DKTK) (Partner Site), Frankfurt, Germany

tumor microenvironments by "shifting" the balance toward a more favorable anti-tumorigenic phenotype. We have also generated a mathematical model to explain the mechanism of action of PD-L1 blockade when combined with RT.

# Results

## PD-L1 expression is upregulated after radiotherapy and chemotherapy in PDAC cells

We analyzed PD-L1 expression by flow cytometry in murine PDAC cell lines 24 h after RT and gemcitabine treatment (Fig 1A and B, and Appendix Fig S1A and B). Both mean fluorescence intensity (MFI) and percentage of PD-L1$^+$ cells significantly increased in the KPC and Pan02 cell lines compared to the DMSO-treated controls. Similarly, we observed PD-L1 upregulation in the human pancreatic cancer PSN-1 cells upon RT and gemcitabine treatment (Appendix Fig S1C). However, conditioned medium (CM) obtained from KPC and PSN-1 cells 24 h after RT and chemotherapy did not alter PD-L1 expression (Appendix Fig S1D and E).

The signal transducer and activator of transcription (Stat) is an important mediator of PD-L1 signaling (Mowen & David, 2000). Hence, we investigated the effect of JAK/Stat signaling on PD-L1 expression in both KPC and Pan02 cell lines (Fig 1C and D). Consistent with our flow cytometry results, we observed upregulation of PD-L1 following RT and gemcitabine treatment by Western blotting. Upon inhibition of the JAK/Stat using AG490, RT- and gemcitabine-mediated upregulation of PD-L1 was completely abolished (Fig 1C and D). Because AG490 can block both Stat1 and Stat3, we inhibited Stat1 and Stat3 by siRNA and confirmed their downregulation by Western blot (Appendix Fig S1F and G). Subsequent flow cytometry analysis of PD-L1 after treatment showed that inhibition of Stat1 (Fig 1E and F) but not Stat3 (Appendix Fig S1H) decreased PD-L1 expression after RT and gemcitabine treatment in PDAC cells.

IFNγ has been shown to stimulate PD-L1 expression downstream of Stat1 in several cancer cells (Dovedi et al, 2014; Kharma et al, 2014). Hence, we examined whether IFNγ expression in KPC and Pan02 cells were affected by treatments (Appendix Fig S2A). Gemcitabine stimulated a small but significant increase in MFI and percentage of IFNγ-expressing cells compared to controls, but RT had no effect. Moreover, IFNγ expression was not altered following Stat1 and Stat3 inhibition in PDAC cells after RT and gemcitabine treatment (Appendix Fig S2B and C). We next asked whether IL-6 affects PD-L1 and IFNγ expression in PDAC cells as IL-6 also signals via the JAK/Stat pathway. Addition of recombinant murine IL-6 did not alter the percentage of PD-L1$^+$ cells (Appendix Fig S3A) or IFNγ expression (Appendix Fig S3B) compared to untreated PDAC cells. These results indicate that RT and gemcitabine treatments can induce PD-L1 expression in PDAC cells in vitro in a JAK/Stat1 but not Stat3-dependent manner and that blockade of JAK/Stat1 signaling can abrogate PD-L1 induction.

## Efficacy of PD-L1 blockade in radiosensitizing PDAC tumor to low radiation doses

Next, we investigated the effect of anti-PD-L1 on tumor cell growth in vitro. Treatment with anti-PD-L1 did not sensitize KPC or Pan02

tumor cells to RT or gemcitabine (Appendix Fig S4A and B). In vivo, KPC syngeneic tumor-bearing mice were treated with either RT alone (6 Gy or 5 × 2 Gy), anti-PD-L1 alone, or their combination (Fig 2A). Treatment with anti-PD-L1 alone had no significant effect on tumor growth, whereas RT radiosensitized tumors compared to untreated controls. Addition of anti-PD-L1 to either RT schedule failed to significantly enhance tumor response to RT (Fig 2A). Similar to the KPC syngeneic tumor model, treatment of Pan02 allografts with either RT (6 Gy or 5 × 2 Gy), anti-PD-L1, or their combination failed to show significant growth delay compared to the untreated control group (data not shown).

We conducted flow cytometry analysis of immune effector CD45$^+$CD8$^+$ T cells, CD45$^+$CD4$^+$ T cells, and CD11b$^+$Gr1$^+$ myeloid cells in the tumor 5 days after completion of the above treatments. We failed to detect any significant changes in any of the immune cell populations in the tumor after the different treatments (Fig 2B and Appendix Fig S5). Treatment was well tolerated with no weight loss observed in any of the mice throughout the experiment. Thus, PD-L1 blockade did not significantly sensitize PDAC tumors to low RT doses (6 Gy or 5 × 2 Gy). Although a trend toward radiosensitization was observed, it is important to note that low RT doses did not increase cytotoxic CD8$^+$ T-cell numbers in the tumor microenvironment.

## PD-L1 blockade sensitizes pancreatic allografts to high RT doses

Because anti-PD-L1 treatment failed to enhance tumor response to relatively low RT doses, we then asked whether combination with higher RT doses could induce a better response to PD-L1 blockade. We first tested the efficacy of high RT doses with anti-PD-L1 in the Pan02 allograft tumor model (Fig 3A). Both high RT doses (12 Gy or 5 × 3 Gy) resulted in significant tumor growth delay, whereas anti-PD-L1 alone did not affect tumor growth. However, PD-L1 blockade significantly enhanced response to both RT schedules. The in vivo experiment had to be terminated at around days 35–39 because the extensive regression seen in several tumors in both combination groups resulted in large wounds that could compromise animal welfare. Flow cytometry analysis revealed significantly increased CD45$^+$CD8$^+$ and CD45$^+$CD4$^+$ T-cell infiltration in irradiated tumors compared to controls; these numbers were additionally significantly increased upon blockade of PD-L1 (Fig 3B and Appendix Fig S6). Compared to the control, CD11b$^+$Gr1$^+$ myeloid cell numbers only decreased significantly after 12 Gy + PD-L1 blockade.

Similar to Pan02, mice bearing KPC tumors were treated with either 12 Gy, 5 × 3 Gy, anti-PD-L1, or the combinations (Fig 4A and B). Both RT doses resulted in increased tumor growth delay that was further enhanced after administration of anti-PD-L1. In parallel, we determined the effect of CD8$^+$ T-cell depletion using anti-CD8 antibodies on the radiosensitization potential of PD-L1 blockade in the KPC model (Fig 4A and B). Of note, the control, the anti-PD-L1 and anti-CD8 alone group are the same in both Fig 4A and B. Treatment with anti-CD8 did not alter KPC tumor growth in either unirradiated or irradiated mice. However, addition of anti-CD8 reversed the radiosensitizing effect of PD-L1 blockade (Fig 4A and B), underscoring the importance of CD8$^+$ T cells in mediating the radiosensitizing effect of PD-L1 blockade in PDAC. As in the Pan02 syngeneic models, CD45$^+$CD8$^+$ and CD45$^+$CD4$^+$ T-cell infiltration significantly increased after irradiation and was further enhanced after

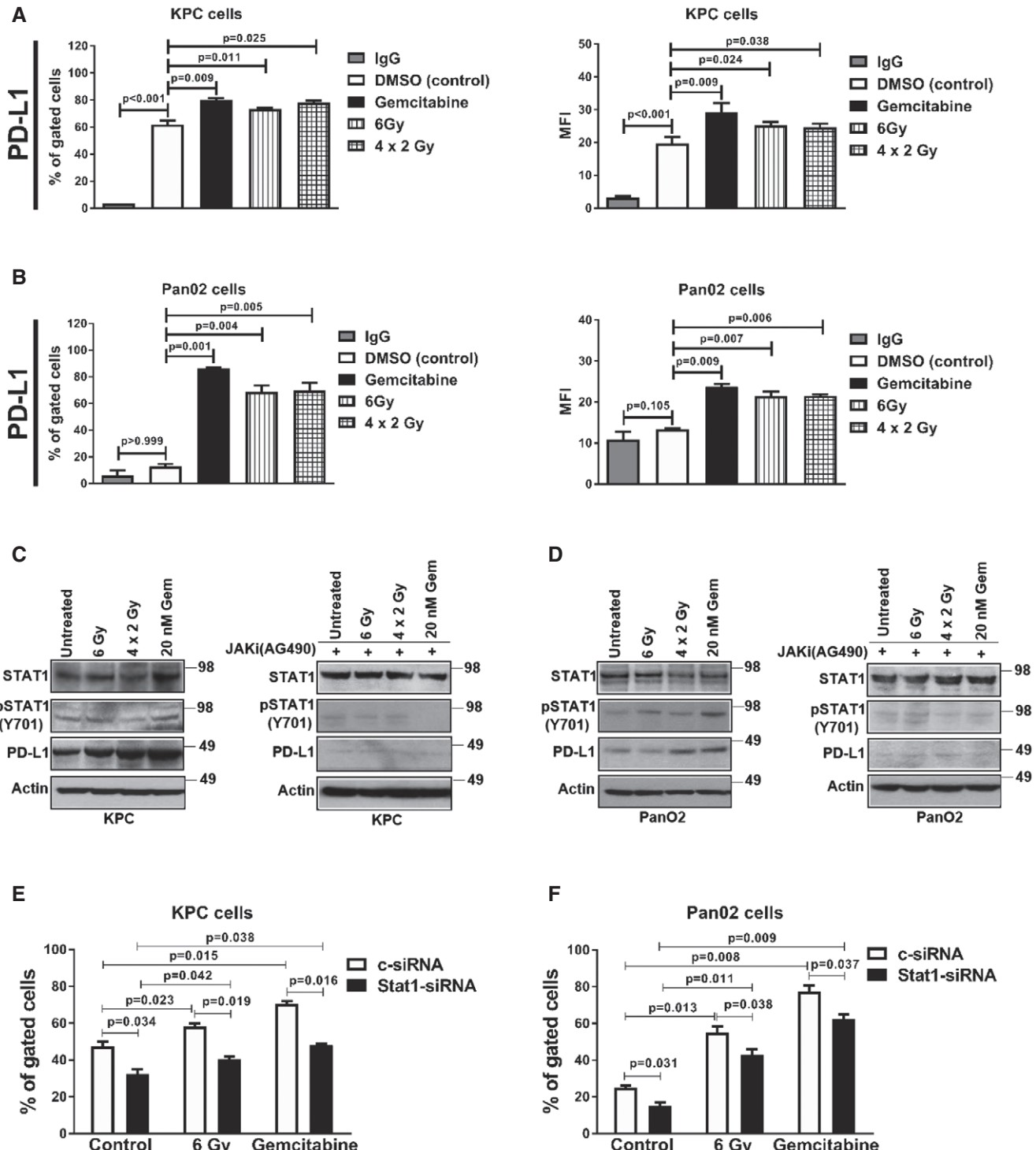

**Figure 1. RT and chemotherapy alter PD-L1 expression in PDAC cells.**

A, B   Flow cytometric analysis of PD-L1 expression in (A) KPC and (B) Pan02 cells following treatment with gemcitabine and RT. Gemcitabine was diluted in DMSO (vehicle) and DMSO was used for the "untreated" control group (mean ± SD, *n* = 3, one-way ANOVA, Bonferroni test). MFI, mean fluorescence index.

C, D   Western blot analysis of the indicated proteins in both KPC (C) and Pan02 (D) cells following radiation and chemotherapy ± AG490, a JAK/Stat kinase inhibitor. Actin represents loading control.

E, F   Flow cytometric analysis of PD-L1 after gemcitabine and RT (as described above) in KPC (E) and Pan02 (F) following Stat1 downregulation by siRNA (mean ± SD, *n* = 3, Student's *t*-test).

Source data are available online for this figure.

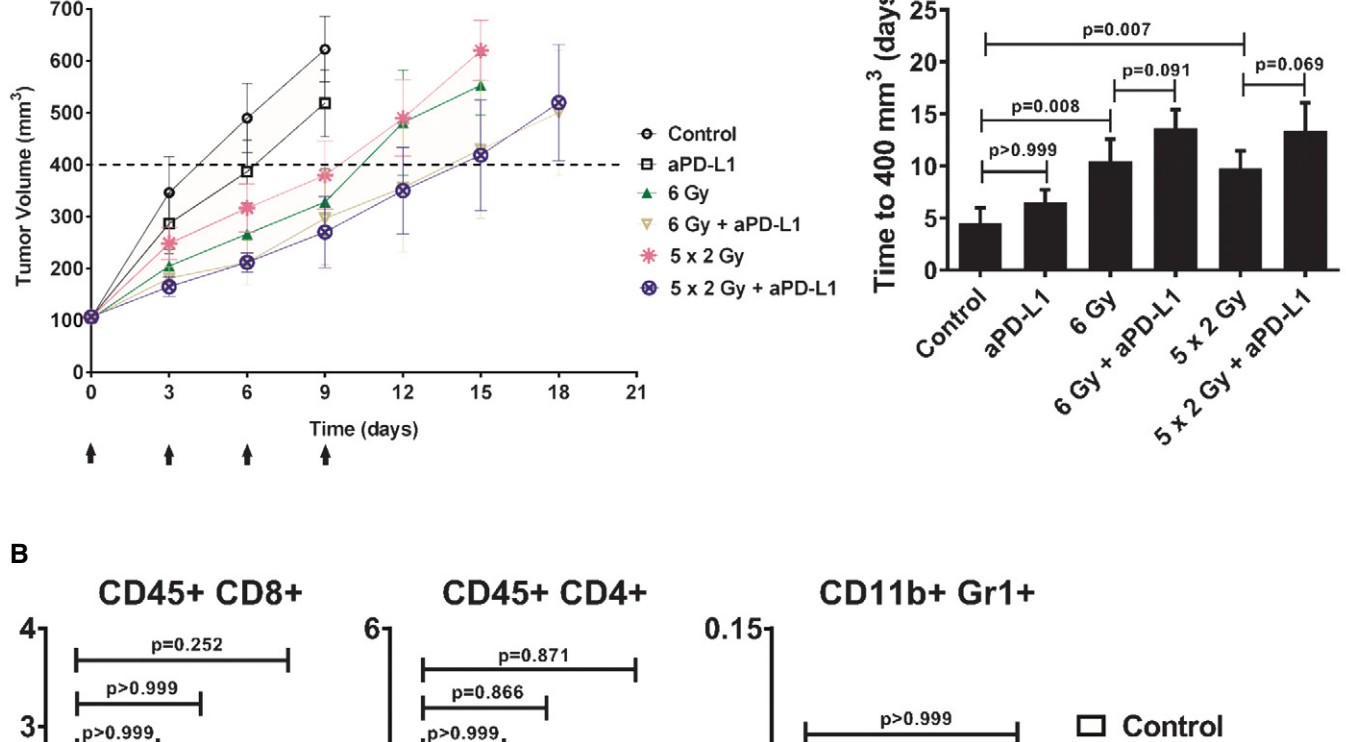

Figure 2. PD-L1 blockade does not significantly sensitize PDAC to low RT doses (6 Gy or 5 × 2 Gy; initiated at day 0) in the syngeneic KPC tumor allograft model.

A   Tumor growth delay was measured in the different groups, as indicated ($n$ = 6 mice per group). Anti-PD-L1 was given at days 0, 3, 6, and 9 (black arrows). The average time (days) for tumors to reach a volume of 400 mm$^3$ from day 0 is shown (means ± SD, $n$ = 1, one-way ANOVA, Bonferroni test). No weight loss was observed in the *in vivo* experiment.

B   Quantitative data on the percentage of gated cells are shown ($n$ = 5 per group; means ± SD, $n$ = 1, one-way ANOVA, Bonferroni test).

PD-L1 blockade (data not shown). Additionally, we assessed the activation status of the CD45$^+$CD8$^+$ T cells based on IFNγ expression and found increased activated CD8$^+$ T cells following combination of high RT doses and PD-L1 blockade (described in Appendix Results and Appendix Fig S7).

We next compared simultaneous combination of 12 Gy with anti-PD-L1 to administration of anti-PD-L1 1 week after RT (Fig 4C). In contrast to the simultaneous combination, sequential administration of anti-PD-L1 1 week post-RT did not radiosensitize KPC tumor allografts. Moreover, we analyzed the effect of PD-L1 blockade after a very high single RT dose (20 Gy) in the KPC model. PD-L1 blockade significantly radiosensitized tumors after 20 Gy, but mice in both the RT alone and the RT + anti-PD-L1 groups developed grade

2 radiation dermatitis that forced termination of the experiment at approximately day 35 (Appendix Fig S9A). Taken together, anti-PD-L1 treatment resulted in significant tumor growth delay after high RT doses that correlated with enhanced tumor infiltration of CD8$^+$ T cells and decreased CD11b$^+$Gr1$^+$ myeloid cells.

### Changes in cytokine profiles after RT and PD-L1 blockade

We examined expression of several inflammatory cytokines in sera of mice after treatment with anti-PD-L1 and/or RT (Appendix Fig S8A). Levels of stromal derived factor 1 (SDF-1) and IL-1 receptor agonist (ra) decreased slightly after anti-PD-L1 and RT treatments in the cytokine array (Appendix Fig S8B). SDF-1 levels were

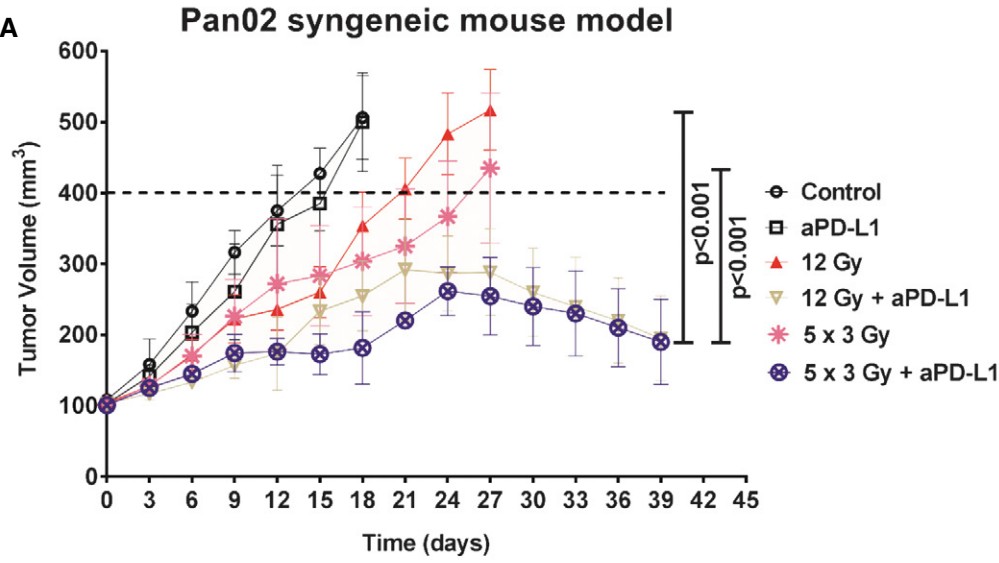

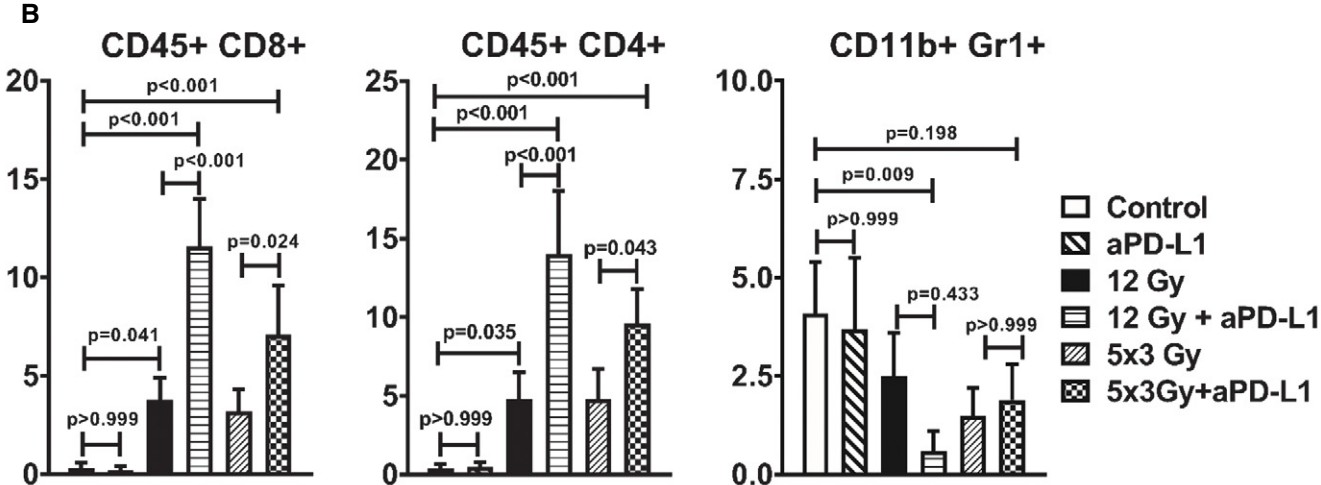

**Figure 3.  RT combined with blockade of PD-L1 amplifies anti-tumor effects in Pan02 tumor allograft.**

A  Tumor growth delay was measured by treating tumors with RT (12 Gy or 5 × 3 Gy daily; initiated at day 0) and anti-PD-L1 alone or combination of RT (*n* = 6 mice per group). Anti-PD-L1 was given at days 0, 3, 6, and 9. The radiosensitizing effect of anti-PD-L1 after 12 and 5 × 3 Gy was assessed as shown by the capped lines (means ± SD, *n* = 1, one-way ANOVA, Bonferroni test). No weight loss was observed in the *in vivo* experiment.

B  Quantitative data on the percentage of gated cells are shown (*n* = 5 per group; means ± SD, *n* = 1, one-way ANOVA, Bonferroni test).

significantly downregulated following RT and combination anti-PD-L1 + RT compared to controls as shown by ELISA (Appendix Fig S8C).

**PD-L1 blockade improves both response to chemoradiotherapy and radiation**

We examined PD-L1 expression in the syngeneic KPC tumor allografts after RT and gemcitabine treatment. PD-L1 was induced 5 days after treatment with gemcitabine, 12 Gy, 20 Gy, and 5 × 3 Gy (Appendix Fig S9B). Similarly, PD-L1 was upregulated 24 h (short term) as well as 3–7 weeks (long term) after

completion of gemcitabine in the transgenic KPC mice compared to control (Appendix Fig S9C). Hence, similar to the *in vitro* conditions, RT and gemcitabine can upregulate PD-L1 in PDAC *in vivo*.

We then assessed the impact of anti-PD-L1 when combined with gemcitabine and RT (Fig 5A). Neither gemcitabine alone, nor anti-PD-L1 alone affected tumor growth compared to control. The combination of anti-PD-L1 with gemcitabine produced a marginally significant tumor growth delay, whereas gemcitabine and RT had a more pronounced effect. Consistent with our previous findings, anti-PD-L1 significantly enhanced the RT-induced tumor growth delay. Compared to the double combinations, the

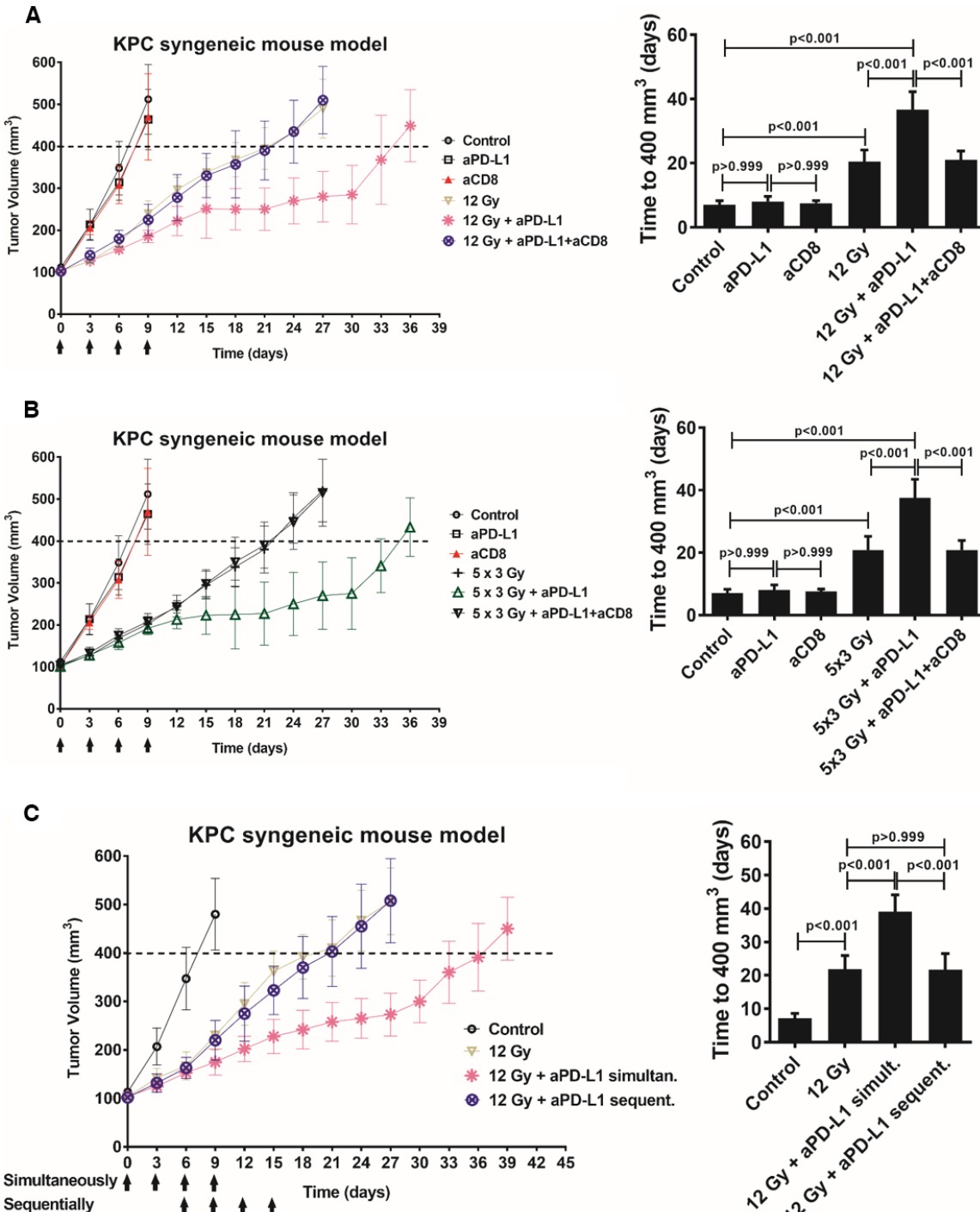

**Figure 4. CD8+ T cells are required for efficacy of RT and anti-PD-L1 treatment.**

A   KPC tumor growth delay after treatment with either 12 Gy (day 0), anti-PD-L1 (days 0, 3, 6, and 9; black arrows), anti-CD8 alone (days 0, 3, 6, and 9) or their combinations, as indicated.

B   KPC tumor growth delay after treatment with either 5 × 3 Gy (days 0–4), anti-PD-L1 alone (days 0, 3, 6, and 9; black arrows), anti-CD8 alone (days 0, 3, 6, and 9) or their combinations, as indicated. Of note, the control, the anti-PD-L1 and anti-CD8 alone groups are the same as in (A).

C   KPC tumor growth delay after either simultaneous (days 0, 3, 6, and 9; black arrows, upper row) or sequential (days 6, 9, 12, and 15; black arrows, lower row) addition of anti-PD-L1 to RT (day 0).

Data information: The average time (days) for tumors to reach a volume of 400 mm³ from day 0 is also shown in the graphs on the right (means ± SD, n = 1, one-way ANOVA, Bonferroni test). No weight loss was observed in the in vivo experiments.

Source data are available online for this figure.

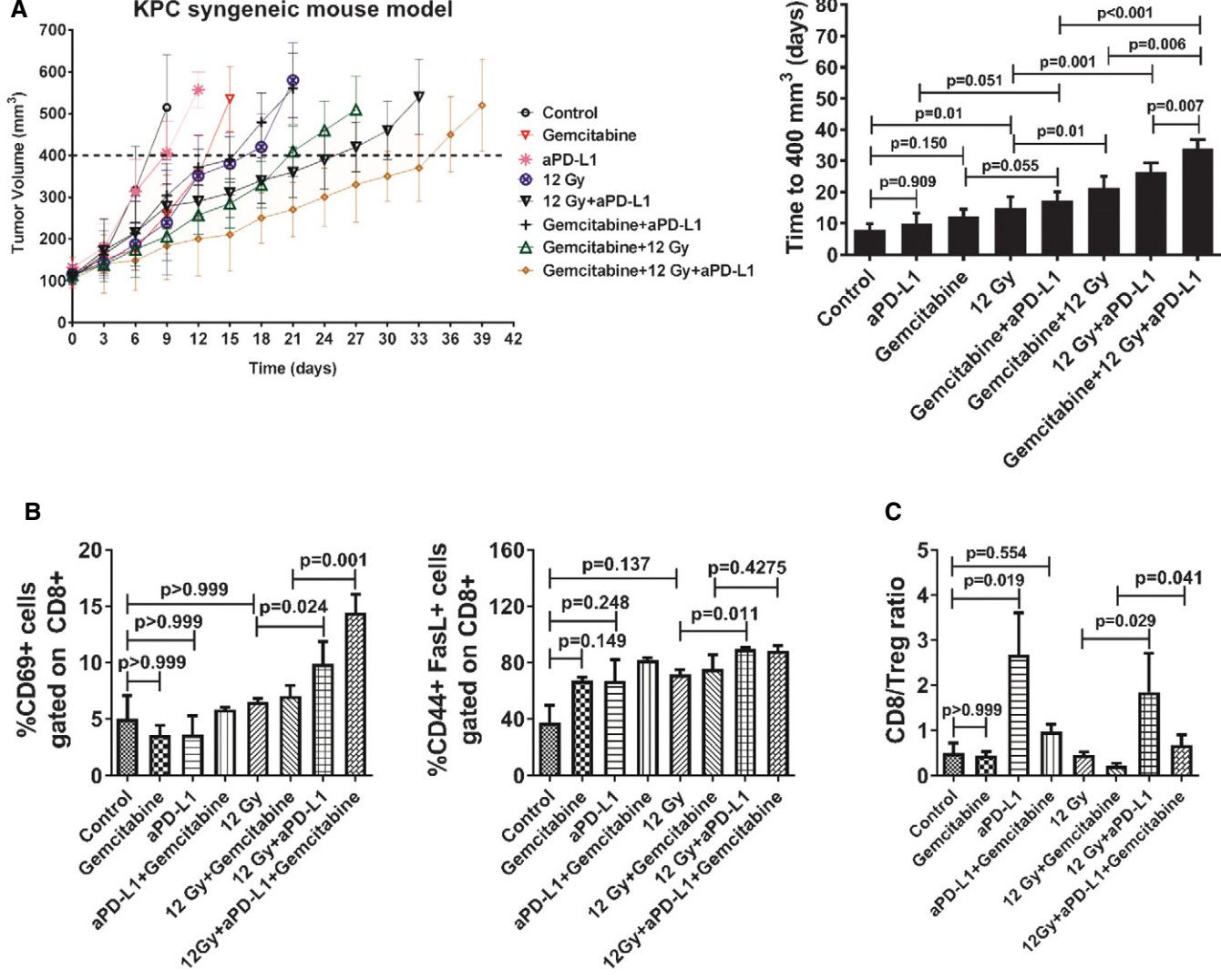

**Figure 5.  Impact of triple combination of gemcitabine-based chemoirradiation with anti-PD-L1 in the KPC tumor allograft.**

A    Mice were treated with either gemcitabine (days 0, 3), anti-PD-L1 (days 4, 7, 10, and 13), 12 Gy (day 4) alone, or in combinations, as indicated (*n* = 8 mice per group). The average time (days) for tumors to reach a volume of 400 mm³ from day 0 is shown (means ± SD, *n* = 1, one-way ANOVA, Bonferroni test). No weight loss was observed in the *in vivo* experiment.

B, C    RT plus anti-PD-L1 reverses T-cell exhaustion (B) and increases CD8:Treg ratio (C). Bar graphs show quantitation of T-cell activation markers and CD8:Treg ratio in tumor samples based on flow cytometry analysis, as indicated (*n* = 5 per group; means ± SD, *n* = 1, one-way ANOVA, Bonferroni test).

triple combination significantly enhanced tumor growth delay, albeit the effect of gemcitabine appeared to be additive rather than synergistic (Fig 5A).

It has been previously reported that T cells become exhausted in cancers and chronic infections (Virgin *et al*, 2009; Herbst *et al*, 2014). Because PD-L1 was upregulated in PDAC cells *in vitro* and *in vivo* following RT and gemcitabine treatment and could potentially suppress T-cell activation, we examined the expression of T-cell activation markers CD69, FasL, and CD44 on intratumoral CD8[+] T cells after treatment with RT, gemcitabine, and/or anti-PD-L1. We did not detect any significant difference in T-cell activation markers after single-agent treatment (Fig 5B). However, addition of anti-PD-L1 to RT and gemcitabine + RT significantly increased numbers of CD69[+] and CD44[+]FasL[+] CD8 T cells compared

to controls. Additionally, treatment with anti-PD-L1 alone, or anti-PD-L1 in combination with RT and RT + gemcitabine increased the CD8/Treg ratio compared to control or treatment with RT alone and RT + gemcitabine, respectively (Fig 5C). These data provide evidence on the potential of PD-L1 blockade to promote T-cell activation in RT, gemcitabine, and combined RT plus gemcitabine.

**Anti-PD-L1 enhances the anti-metastatic effect of RT**

We asked whether anti-PD-L1 could enhance the cytotoxic effect of RT in a microenvironment other than the tumor allografts. For that purpose, we assessed the effect of PD-L1 blockade in liver metastasis. Mice were injected intrasplenically with either naïve or pre-irradiated (12 Gy) KPC cells, then treated

with anti-PD-L1 every 3 days. After 14 days, livers were harvested and weighted. Compared to control, prior treatment with 12 Gy reduced metastatic tumor burden that was further enhanced by anti-PD-L1 treatment (Fig 6A). Combination of 12 Gy + anti-PD-L1 also increased CD8$^+$ T-cell infiltration, while CD11b$^+$Gr1$^+$ myeloid cell and CD4$^+$CD25$^+$FOXP3$^+$ Treg numbers were significantly reduced compared to controls (Fig 6B and Appendix Fig S10). Hence, the combination of RT and PD-L1 can enhance the cytotoxic effect of RT in an environment different from the primary tumor, further supporting the use of this therapeutic strategy.

## Mathematical modeling of immune cells interacting with tumor cells

Finally, we used mathematical modeling (Dushek *et al*, 2011; Lever *et al*, 2014) to provide additional insight into the tumor growth behavior and underlying mechanism of synergy between RT and anti-PD-L1 (Appendix Fig S11A). For that purpose, we simulated populations of tumor cells, immune cells, and the effects of RT and anti-PD-L1 in different ways (i.e., assuming different mechanisms of

action) and compared the overall tumor growth predicted by these different models to our experimental data.

Experimentally, we observed that RT slowed overall tumor growth for up to 4 weeks, but our initial simulations failed to reproduce this long-term effect if we modeled RT as directly killing tumor cells (Appendix Fig S11B) or reducing their overall growth rate (i.e., permanently decreasing division rate or increasing cell death rate, Appendix Fig S11C). Given that RT is known to induce immune cell infiltration (Burnette & Weichselbaum, 2013), we generated a model where irradiated tumor cells became susceptible to killing by external factors (immune cells) that increase over time. Tumor growth simulations from this model can reproduce the effects of RT observed in our experiments (Appendix Fig S11D).

Building on this model for RT, we next simulated the tumor growth curves for different mechanisms of PD-L1 blockade. We found that two different models can reproduce the synergistic effects observed in our experiments: Anti-PD-L1 increases CD8$^+$ T-cell recruitment (Appendix Fig S11E) or CD8$^+$ T-cell killing efficiency (Appendix Fig S11F). Both mechanisms are supported by our experiments (Figs 3–5) and therefore it is likely that both contribute to tumor control.

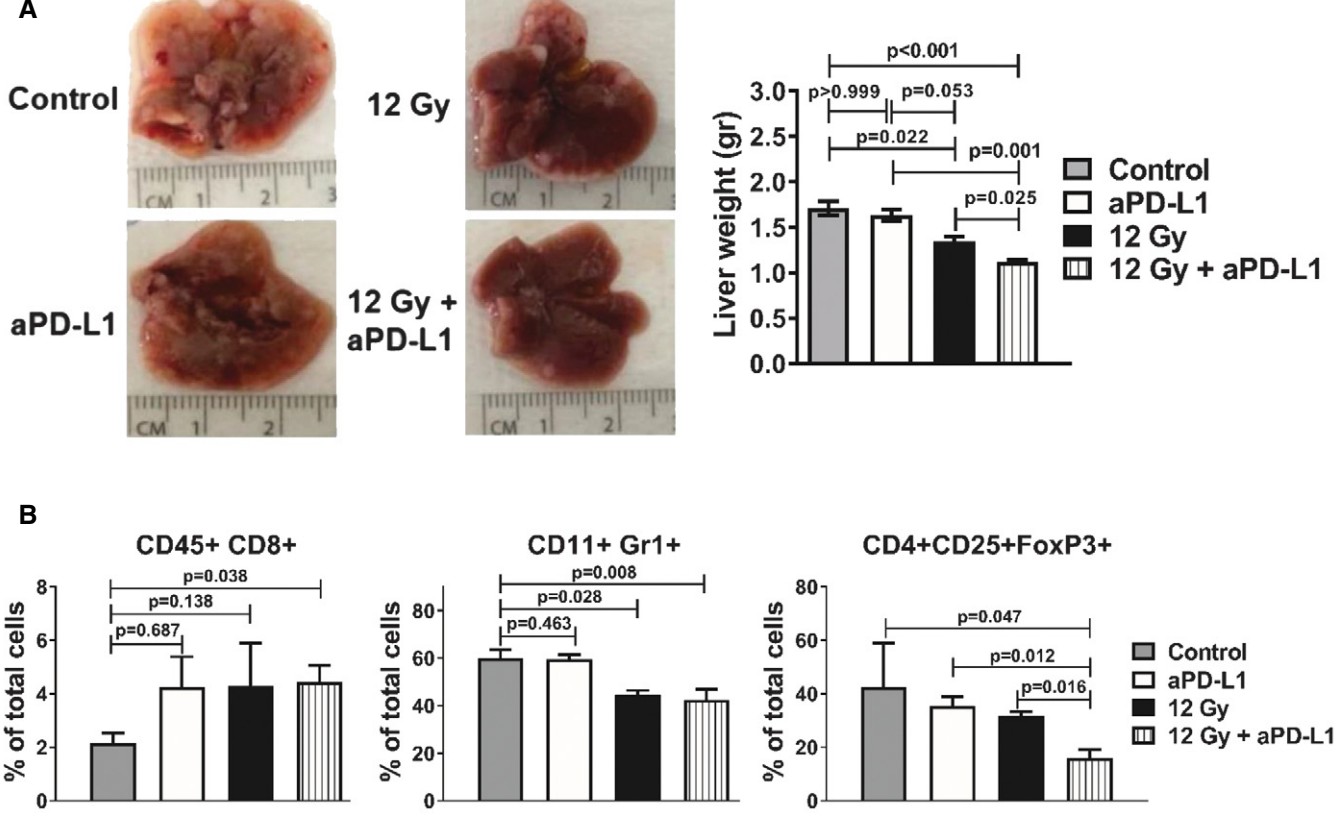

**Figure 6.  Blockade of PD-L1 enhances the anti-metastatic effect of RT.**

A  Representative images of livers from the different treatment groups, as indicated. The corresponding liver weights are shown on the right (*n* = 6 mice per group; means ± SD, *n* = 2, one-way ANOVA, Bonferroni test).

B  Flow cytometry analysis of CD45$^+$CD8$^+$, CD11b$^+$Gr1$^+$, and CD4$^+$CD25$^+$FOXP3$^+$ cell populations in the livers of the mice from the different treatment groups, as indicated (*n* = 5 mice per group; means ± SD, *n* = 1, one-way ANOVA, Bonferroni test).

Lastly, we observed that radiosensitization was completely abolished when anti-PD-L1 was administered 7 days after RT (Fig 4C). The model could reproduce this result if it included suppressive microenvironment factors established earlier than 7 days (Appendix Fig S11G). The administration of anti-PD-L1 early but not late shifts the balance to a more inflammatory environment by preventing the establishment of factors that suppress the ability of CD8[+] T cells to kill tumor cells (Appendix Fig S11G and E). This is in line with our findings of reduced CD11b[+]Gr1[+] and Treg numbers in tumors treated with anti-PD-L1 and RT (Figs 3–6).

In summary, we find that the synergy between RT and anti-PD-L1 observed in our experimental tumor growth curves can be explained by a model where (i) RT induces a large population of tumor cells to become susceptible to immune cell killing and (ii) anti-PD-L1 increases the recruitment and killing ability of CD8 T cells and, importantly, prevents the establishment of suppressive microenvironment factors (Appendix Fig S11A and G).

## Discussion

PD-L1 plays a central role in permitting cancer evasion, mainly by interfering with T-cell functions, and inhibitors of the PD-1/PD-L1 axis have changed the paradigm in the management of melanoma patients (Clark *et al*, 2007). However, PDAC is characterized by a highly immunosuppressive stroma (Hiraoka *et al*, 2006; Lutz *et al*, 2014; Beatty *et al*, 2015; Diana *et al*, 2016b; Zhang *et al*, 2017) and immune checkpoint inhibitors alone were ineffective in the clinical setting (Royal *et al*, 2010; Brahmer *et al*, 2012). Evidence indicates potentially complementary roles for immunotherapy and RT (Sharabi *et al*, 2015a), but this combination has not been explored in PDAC. Here, we show that blockade of PD-L1 strongly enhanced tumor response to high (12, 5 × 3, and 20 Gy) but not low (6 and 5 × 2 Gy) RT doses, albeit a trend was noted for low doses. In addition to sensitizing tumors to RT, anti-PD-L1 improved response after gemcitabine-based chemoradiation. This is, to our knowledge, the first preclinical study to examine in detail the impact of this therapeutic strategy in PDAC. Our findings support and build on previous preclinical reports demonstrating enhanced radiosensitization using CTLA-4 and/or PD-1/PD-L1 immune checkpoint inhibitors in melanoma, breast, and colorectal cancer, and glioblastoma multiforme (Demaria *et al*, 2005; Deng *et al*, 2014; Dovedi *et al*, 2014; Sharabi *et al*, 2015b; Twyman-Saint Victor *et al*, 2015; Rodriguez-Ruiz *et al*, 2016).

Although our observations with higher RT doses are in agreement with the majority of reports showing strong radiosensitization with PD-1/PD-L1 (Demaria *et al*, 2005; Deng *et al*, 2014; Dovedi *et al*, 2014; Sharabi *et al*, 2015b; Rodriguez-Ruiz *et al*, 2016) and CTLA-4 blockade (Twyman-Saint Victor *et al*, 2015), some suggestions that the effects of immunity could be overcome were noted; despite the significant growth delay after combination RT + anti-PD-L1, tumors eventually regrew in the KPC model. Unfortunately, the *in vivo* experiment using the Pan02 model had to be terminated early due to the profound tumor regression that led to the formation of extensive open wounds and hence tumor regrowth could not be monitored in this model. Also, 20 Gy treatment in KPC allografts resulted in RT-induced dermatitis (also in the RT alone group) that made tumor rechallenge impossible as the experiment had to be terminated.

Interestingly, PD-L1 exerted a radiosensitizing effect only with higher RT doses in our models. In contrast, Dovedi *et al* demonstrated that PD-L1 inhibition in combination with lower RT doses (5 × 2 Gy) was sufficient to delay tumor growth in colorectal and triple-negative breast cancers (Dovedi *et al*, 2014). These differences could be attributed to the cell line behavior, such as higher intrinsic radiosensitivity compared to PDAC cells (Liu *et al*, 2011; Tolaney *et al*, 2015) that could render these cells more vulnerable to recognition and cytotoxic killing by T cells.

PD-L1 expression is upregulated by either oncogenic signaling, such as PI3K/AKT activation (innate resistance) or inflammatory processes via IFNγ (adaptive resistance) (Parsa *et al*, 2007; Kharma *et al*, 2014). In accordance with other groups (Spivak-Kroizman *et al*, 2013; Deng *et al*, 2014; Dovedi *et al*, 2014; Peng *et al*, 2015), we showed that both RT and gemcitabine induced PD-L1 expression in PDAC both *in vitro* and *in vivo*. Upregulation of PD-L1 by RT and gemcitabine lends support to coupling PD-L1 blockade with these conventional treatments to enhance anti-tumor response. Furthermore, PD-L1 expression can be suppressed by depletion of Stat1 (Kharma *et al*, 2014) and we found that the JAK/Stat kinase inhibitor AG490 abrogated baseline PD-L1 expression and upregulation after conventional treatments in our series. Because AG490 can inhibit both Stat1 and Stat3, we also used siRNA to demonstrate that inhibition of Stat1 but not Stat3 decreased PD-L1 expression at baseline, and after RT and gemcitabine treatment in PDAC cells. Interestingly, IL-6 had marginal effects on PD-L1 expression in the PDAC cells despite activating the JAK/Stat pathway and its previous implication in PD-L1-mediated immunosuppression (Spary *et al*, 2014; Chen *et al*, 2016).

PD-L1 upregulation can induce dysfunction in T cells leading to their exhaustion and impaired activation and immune surveillance (Wherry, 2011). Because both RT and gemcitabine upregulated PD-L1 in PDAC, we assessed whether this suppressed CD8[+] T-cell activation. Addition of anti-PD-L1 to RT and RT + gemcitabine increased the proportion of intratumoral CD8[+] T cells expressing the activation markers CD69, CD44, and FasL (Smith-Garvin *et al*, 2009; Wherry, 2011; Twyman-Saint Victor *et al*, 2015). These data suggest that despite inducing PD-L1 expression in PDAC, addition of anti-PD-L1 to RT and gemcitabine can overcome T-cell suppression and enhance T-cell activation.

Radiotherapy can modulate the immune system to promote CD8[+] T cells to attack tumor cells through multiple mechanisms (Burnette & Weichselbaum, 2013; Sharabi *et al*, 2015a) including the release of stimulatory cytokines such as IFNγ. We only observed upregulation of IFNγ expression in PDAC cells after treatment with gemcitabine but not with RT. IFNγ can activate Jak/Stat resulting in PD-L1 upregulation (Bellucci *et al*, 2015), which could explain PD-L1 upregulation in PDAC cells after gemcitabine treatment. In an attempt to uncover other cytokines involved in the RT-mediated immune response, we examined the serum cytokine profiles of mice after treatment. RT alone or in combination with anti-PD-L1 significantly decreased SDF-1 expression. Interestingly, SDF-1 has been previously implicated in immunosuppression and tumor progression in PDAC (Feig *et al*, 2013) and downregulation of SDF-1 may contribute to the therapeutic effect of RT + anti-PD-L1 in reversing the immunosuppressive phenotype.

The positive prognostic role of high CD8[+] T-cell infiltration has been demonstrated consistently in the clinical setting in several

malignancies, including PDAC (Fridman *et al*, 2012; Ino *et al*, 2013; Diana *et al*, 2016a). We observed increased CD8$^+$ T-cell infiltration in tumors after high RT doses that was further enhanced by inhibition of PD-L1. Anti-PD-L1 alone and in combination with RT significantly increased the CD8/Treg ratio, an important metric of anti-tumor response to treatment (Byrne *et al*, 2011; Amedei *et al*, 2013; Diana *et al*, 2016a). Furthermore, blockade of CD8 abrogated the radiosensitizing effect of anti-PD-L1 in PDAC cells *in vivo*, in agreement with other reports (Deng *et al*, 2014; Dovedi *et al*, 2014; Twyman-Saint Victor *et al*, 2015; Rodriguez-Ruiz *et al*, 2016).

Addition of anti-PD-L1 to RT also caused a reduction in CD11b$^+$Gr1$^+$ myeloid cells *in vivo*. Infiltration of myeloid cells is a key feature of cancer inflammation in PDAC and CD11b$^+$Gr1$^+$ cells characteristic of myeloid-derived suppressor cells (MDSCs) are known to suppress antigen-specific T cells (Song *et al*, 2005). The reduction in CD11b$^+$Gr1$^+$ cells appears to be mutually exclusive from CD8$^+$ T-cell infiltration (Clark *et al*, 2007). Deng *et al* revealed that addition of anti-PD-L1 to RT eliminated MDSCs from the tumor microenvironment (Deng *et al*, 2014). Consistent with this, we found reduced CD11b$^+$Gr1$^+$ cells following anti-PD-L1 and RT treatments.

We additionally assessed whether anti-PD-L1 could enhance radiosensitization of PDAC cells in an environment other than the syngeneic allograft model. For that purpose, we intrasplenically injected untreated or previously irradiated (12 Gy) PDAC cells and assessed the impact of systemic PD-L1 blockade (i.p.) in the development of liver metastasis. Both RT alone and anti-PD-L1 alone reduced metastatic colonization compared to untreated controls, but addition of anti-PD-L1 significantly enhanced the anti-metastatic effect of RT. Recruitment of immune cells to the liver is known to precede development of liver metastases (Engblom *et al*, 2016). We found significant infiltration of CD8$^+$ T cells after RT and anti-PD-L1 treatments compared to control. RT in combination with the anti-PD-L1 also significantly reduced infiltration of CD4$^+$ CD25$^+$FOXP3$^+$ T regulatory cells (Byrne *et al*, 2011), further supporting the notion that PD-L1 checkpoint inhibition together with RT can counteract the immunosuppressive tumor environment. Our observations are in line with a previous report showing decreased lung metastases after RT and CTLA-4 blockade in a mouse model of breast cancer (Demaria *et al*, 2005). Also, albeit in a different setting, combination of local RT with immune checkpoint blockade has demonstrated the so-called abscopal effect, that is, regression of metastatic lesions located outside the radiation field in melanoma patients due to the development of immune surveillance (Postow *et al*, 2012).

Finally, we developed a minimal model of immune cell interactions with tumor cells that can explain the tumor growth behavior under treatment conditions. Importantly, we found that the synergistic effect of anti-PD-L1 treatment with RT cannot be explained solely with increased CD8$^+$ T-cell recruitment or activation. An additional effect of anti-PD-L1 on the immunosuppressive microenvironment had to be invoked to explain the delayed tumor growth. Our model indicates that radiation + anti-PD-L1 combination therapy is only effective in patients with pre-existing CD8 T cells in the tumor microenvironment and that anti-PD-L1 should be administered early, in conjunction with RT for maximal anti-tumor responses. The mathematical model provides validation that

anti-PD-L1 additionally targets the immunosuppressive cells, whereas radiotherapy alone may only stimulate immune cell recruitment. These findings are important for the design of future therapeutic strategies as they underscore the clinical value of combination therapy involving PD-1/PD-L1 blockade to target the immunosuppressive microenvironment.

In summary, PD-L1 blockade can improve PDAC response to RT and chemoradiation and can enhance the effect of RT in preventing formation of liver metastases. Altogether, these findings indicate that RT combined with PD-L1 checkpoint inhibition can "shift" the balance toward a more favorable immune phenotype. Our data add important insight on the potential of immune checkpoint inhibitors to radiosensitize PDAC, a tumor that is traditionally considered as non-immunogenic. These findings provide a rationale for exploring the therapeutic potential of PD-L1 blockade in combination with RT in PDAC and should be explored in future clinical studies in patients with PDAC.

## Materials and Methods

### Cell culture and reagents

The murine PDAC cell lines KPC and Pan02 were cultured in DMEM supplemented with 10% FBS and 1% penicillin/streptomycin. KPC cells were derived from KrasLSL$^{G12D}$/+; p53$^{R172H}$/+; Pdx1-Cretg/+ (KPC) tumors. Murine Pan02 pancreatic adenocarcinoma cells were obtained from the NCI–DCTD Tumor Repository, Maryland, USA. Gemcitabine was purchased from Eli Lilly and Company Ltd, UK. Murine anti-PD-L1 antibody (Clone 10F.9G2) was purchased from BioXCell.

### Clonogenic survival assay

Both KPC and Pan02 cells were seeded into 6-well plates**.** Anti-PD-L1 antibody was added to the culture 1 h before irradiation with single dose of 2, 4, and 6 Gy. Similarly, cells were also treated with gemcitabine (0, 10, 20, and 50 nM). Gemcitabine was diluted in DMSO (vehicle), and hence, we used DMSO for the untreated control group. Twenty-four hours after treatment, cells were washed with PBS and replaced with fresh medium. The clonogenic survival of cells was assessed as described previously (Prevo *et al*, 2008).

### Syngeneic mouse models

C57BL/6 mice were injected subcutaneously (s.c.) with KPC and Pan02 cells ($5 \times 10^5$ cells/100 µl in serum-free medium). Tumor size was measured with caliper using the formula Volume = $(a \times b^2)/2$, in which *a* and *b* are the largest and the smallest perpendicular diameters, respectively. Mice were randomized into different groups when tumors reached 100 mm$^3$ and treatment was initiated (day 0) as described previously (Fokas *et al*, 2012). RT was initiated at day 0 (with the exception of the triple combination experiment where it was initiated at day 4) and was administered at a dose of either 6 Gy, 12 Gy, 20 Gy, 5 × 3 Gy, or 5 × 2 Gy given daily using the Gulmay 320 irradiator under inhalational anesthesia with isoflurane (2%). Only the tumor was irradiated, whereas the

rest of the body was sealed to avoid unwanted side effects (Fokas *et al*, 2012). Gemcitabine (100 mg/kg) was given by intraperitoneal (i.p.) injection at days 0 and 3. Anti-PD-L1 antibody (clone 10F.9G2, BioXCell, UK) was administered at a dose of 10 mg/kg by i.p. injection at days 0, 3, 6, and 9 when combined with RT, or at days 4, 7, 10, and 13 in the triple combination experiment (gemcitabine + anti-PD-L1 + RT). For the combination doses, anti-PD-L1 was administered immediately after RT. The anti-CD8 (clone 2.43, BioXCell, UK) to deplete $CD8^+$ T cells was given by i.p. injection at dose of 250 µg at days 0 (immediately post-RT), 3, 6, and 9. We assessed the time it took for the tumors to grow to 400 $mm^3$, starting from the beginning of treatment (day 0) to ensure that our experiments were according to the UK Home Office guidelines for animal welfare.

### Liver metastasis mouse model

To generate liver metastasis, KPC cells ($5 \times 10^5$ cells/100 µl in PBS) were injected intrasplenically into C57BL/6 mice that are either naïve or pre-irradiated with 12 Gy 1 h before intrasplenic injection, as previously described (Lim *et al*, 2015). Spleen was removed 1 min following intrasplenic injection to avoid tumor growth in the spleen. Similar to the allograft experiments, anti-PD-L1 antibody was given at days 0, 3, 6, and 9 by i.p. injection at a dose of 10 mg/kg. Livers were harvested and weighed at day 14.

### KPC transgenic mice

The KPC transgenic mouse model has been described previously (Hingorani *et al*, 2005). When mice reached an age of 11–12 weeks, monitoring for tumor was initiated using palpation twice per week and sonography (Ultrasound-Vevo 700). Treatment with gemcitabine (100 mg/kg) by i.p. injection for a total of three doses every 4 days was started when tumor reached 80–120 $mm^3$. We assessed the impact of chemotherapy either short term (24 h) or long term (3–7 weeks, based on tumor size and deterioration of health status including weight loss > 20% and lethargy) after completion of treatment with gemcitabine. Tumors were harvested and analyzed for PD-L1 expression by flow cytometry as described below.

All animal experiments were performed according to the regulations of the University of Oxford and the Home office, UK (Scientific Procedure Act, 1986; Project License Number 30/2922 issued by the Home Office). All protocols were approved by the Committee on the Ethics of Animal Experiments of the University of Oxford.

### Flow cytometry analysis

KPC and Pan02 cells were seeded in 6-well plate. After 24 h, cells were treated with either 6, $4 \times 2$ Gy fractionated radiation given daily using cesium-137 irradiators, or 20 nM gemcitabine. Gemcitabine was diluted in DMSO (vehicle), and hence, we used DMSO for the untreated control group and the radiation groups as well. Cells were harvested 24 h post-treatment and washed with cold PBS. Cells were stained with mouse PE-PD-L1 and APC-IFNγ. All the antibodies were purchased from eBiosciences.

We also examined expression of PD-L1 following blockade of either Stat1 or Stat3 using siRNA. For that purpose, cells were transfected with 50 nM negative control siRNA (catalog number 1022076, Qiagen), 50 and 100 nM Stat1 (J-058881-10, DermaFECT), or 50 nM Stat3 siRNA (J-040794-10, DermaFECT) using Lipofectamine 2000 (Invitrogen) reagent. After 48 h, proteins were isolated and knockdown was confirmed by Western blotting. For flow cytometry analysis of PD-L1 and IFNγ, siRNA-transfected cells were either irradiated with 6 Gy or treated with gemcitabine and PD-L1 expression examined as above. PD-L1 expression was also assessed following addition of recombinant mouse IL-6 (50 ng/ml; PeproTech) for 1 h and subsequent treatments with 6 Gy and gemcitabine for 24 h.

For flow cytometry analysis of mouse tissue samples, tumor, liver, and spleen were cut into small pieces and only liver samples were digested with 0.05% collagenase/dispase (Roche) for 15 min at 37°C in the presence of 0.01% trypsin inhibitor. Bone marrow was collected by flushing both femur and tibia with PBS using a 25-G syringe. All cells were washed with cold PBS. Cells were then filtered through 70-µm nylon strainer followed by red blood cell (RBC) lysis for 3 min at room temperature. Single-cell suspensions were then stained with the following antibodies: FITC-CD45, FITC-CD8, PeCy7-CD4, PeCy7-CD11b, PE-Gr1, PE-Ly6G and APC-IFNγ, PE-CD25, APC-FOXP3, CD69-PE, APC-CD44, and PeCy7-FasL.

Blood samples were collected in 40 mM EDTA and mixed with equal volume of PBS. Diluted blood was layered onto an equal volume of Histopaque (Sigma) and centrifuged at 400 *g* for 20 min at RT to isolate leukocyte population. The cells were washed with cold PBS and stained with FITC-CD8, PeCy7-CD4, and PE-PD-1. Flow cytometry was performed using a FACSCalibur (Becton Dickinson) and analyzed using FlowJo software (FlowJo, LLC, USA). All the antibodies were purchased from eBiosciences, UK.

### Western blotting

Cells were first treated with 20 µM of JAK kinase inhibitor (AG490, Tocris) 1 h before RT (6 Gy) or addition of 20 nM gemcitabine. Following 24 h post-RT and gemcitabine, cell lysates were prepared in RIPA buffer containing protease inhibitor cocktail (Roche) and phosphatase inhibitor cocktail (Roche). In total, 30 µg of proteins were separated on SDS–PAGE (Invitrogen) and transferred to Hybond-C membranes (Amersham Biosciences). Standard procedures were applied for Western blotting. The following antibodies were used: Stat1, phospho-Stat1 (Y701), Stat3, mouse PD-L1 (all from R&D Systems), actin (Cell Signaling).

### Cytokine profile and SDF-1 ELISA in KPC allografts

Serum was obtained from the KPC allograft-bearing mice 5 days after treatment with either vehicle (control), 12 Gy, anti-PD-L1, or 12 Gy + anti-PD-L1. Serum was assessed for expression of cytokines using a standard Cytokine Profiler kit and SDF-1 ELISA kit (both from R&D Systems) according to the manufacturer instructions. Array spot intensities were quantified using a Matlab software as reported (Zhao *et al*, 2013).

### Mathematical modeling

We and others have previously used mathematical modeling to describe T-cell activation and antigen potency (Dushek *et al*, 2011;

**The paper explained**

**Problem**

Pancreatic ductal adenocarcinoma is characterized by a dismal prognosis and it is refractory to conventional treatment, such as radiotherapy and chemotherapy, and immune checkpoint inhibitor monotherapy. In contrast to the notion that all patients with PDAC die of metastatic disease, at least 20% suffer and die of local recurrence. Hence, there is a need to improve radiotherapy efficacy to enhance local control and clinical outcome. Evidence indicates potentially complementary roles for immunotherapy and radiotherapy, but this combination has not been explored in PDAC.

**Results**

Here, we show that radiation and gemcitabine chemotherapy have substantial immunomodulatory effects, including upregulation of PD-L1 *in vitro* and *in vivo*. Inhibition of PD-L1 strongly enhanced tumor response to high (12 Gy, 5 × 3 Gy, 20 Gy) but not low (6 Gy, 5 × 2 Gy) RT doses, and to gemcitabine-based chemoradiotherapy in different mouse models of PDAC. The increased tumor response to radiation observed upon PD-L1 blockade correlated with increased intratumoral T-cell numbers and activation and reduced myeloid cell infiltration, whereas depletion of CD8$^+$ T cells abrogated the radiosensitization by anti-PD-L1. Additionally, inhibition of PD-L1 enhanced the impact of radiotherapy in preventing development of liver metastases. We generated a novel mathematical model that explains the synergistic effect of radiation and PD-L1.

**Impact**

In the present work, we provide a strong rationale for testing the use of PD-L1 immune checkpoint inhibition with radiotherapy in PDAC in the clinical setting that could lead to improved outcome in patients with this disease.

Lever *et al*, 2014; Tolaney *et al*, 2015). Here, we used the same principles to model the efficacy of anti-PD-L1 and RT in the different experiments. The mathematical model consists of a system of ordinary differential equations (ODEs), which are often used to describe the interactions among cellular populations in immunology and cancer,

| | |
|---|---|
| $\partial T1/\partial t = \lambda * T1$ | (resistant tumor cells) |
| $\partial T2/\partial t = \lambda * T2 - \varepsilon 1/((K+M))*E*T2 - \varepsilon 2*X*T2$ | (susceptible tumor cells) |
| $\partial E/\partial t = \gamma$ | (CD8 effector T cells) |
| $\partial X/\partial t = \beta$ | (other immune cells) |
| $\partial M/\partial t = \varphi$ | (microenvironment factors) |

where: $\lambda = 0.19$, $\gamma = 0.025$, $\beta = 0.16$, $\varepsilon 2 = 0.0298$, $M \geq 1$, $K \ll 1$, tumor volume $V = T1 + T2$, and under the different treatment conditions: no treatment: $\varepsilon 1 = 0.002$, $\varphi = 3$, T1 day 0 = 100, T2 day 0 = 0; radiation: $\varepsilon 1 = 0.002$, $\varphi = 3$, T1 day 0 = 0.4, T2 day 0 = 99.6; radiation + anti-PD-L1 (day 0): $\varepsilon 1 = 0.02$, $\varphi = -0.2$, T1 day 0 = 0.4, T2 day 0 = 99.6; radiation + anti-PD-L1 (day 6): $\varepsilon 1$ day 0 = 0.002, $\varepsilon 1$ day 6 = 0.02, $\varphi$ day 0 = 3, $\varphi$ day 6 = −0.2, T1 day 0 = 0.4, T2 day 0 = 99.6. The ODEs were integrated in Matlab (Mathworks, MA, USA) using ode45.

**Statistical analyses**

Differences between groups were determined using Student's paired *t*-test and one-way ANOVA (Bonferroni test) with the GraphPad Prism version 5 (GraphPad software, USA). *P*-values lower than 0.05 were considered statistically significant. Error bars represent standard deviation.

**Expanded View** for this article is available online.

### Acknowledgements

This work was funded by CRUK, the Marie Curie Innovative Training Network and the Kidani Memorial Trust. This work was partly funded by a Sir Henry Dale Fellowship jointly funded by the Wellcome Trust and Royal Society (grant 098363). We thank Dr. Ana Gomez, Mick Woodcock, and Graham Brown for the technical support.

### Author contributions

AA, SYL, and EF designed the experimental strategy. AA, SYL, ZD'C, and EF organized the experiments and collected and analyzed the data. AD and SYL provided assistance with the experimental methodology. AA, SYL, and EF prepared the protocols for mouse radiotherapy. OJS, WGM, and RJM provided experimental advice. PK and OD generated the mathematical modeling. AA, SYL, and EF wrote the manuscript with advice from OJS, WGM, OD, and RJM. KJ assisted with the *in vivo* studies. SL helped with the manuscript preparation. All authors discussed and approved the manuscript.

### Conflict of interest

Part of this work was presented at the ESTRO 35 and the NCRI 2015 conference. The authors declare that they have no conflict of interest.

### For more information

https://www.ncbi.nlm.nih.gov/pubmed/26433823
https://www.ncbi.nlm.nih.gov/pubmed/26270858

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
