## [Review Process File · EMBO Molecular Medicine]

PD-L1 blockade enhances response of pancreatic ductal adenocarcinoma to radiotherapy

Abul Azad, Su Yin Lim, Zenobia DiCosta, Keaton Jones, Angela Diana, Owen J. Sansom, Philipp Kruger, Stanley Liu, W. Gillies McKenna, Omer Dushek, Ruth J. Muschel, and Emmanouil Fokas

Corresponding author: Emmanouil Fokas, University of Oxford

Review timeline:

Submission date:	06 June 2016
Editorial Decision:	05 July 2016
Revision received:	04 October 2016
Editorial Decision:	19 October 2016
Revision received:	26 October 2016
Accepted:	28 October 2016

Transaction Report:

Editor: Roberto Buccione

1st Editorial Decision

05 July 2016

Thank you for the submission of your manuscript to EMBO Molecular Medicine. We have now heard back from the Reviewers whom we asked to evaluate your manuscript.

We are sorry that it has taken longer than usual to get back to you on your manuscript. In this case we experienced difficulties in securing appropriate reviewers and then obtaining their evaluations in a timely manner. Further to this, I wished to discuss the evaluations further with my colleagues

As you will see, while Reviewer 2 is more positive, Reviewers 1 and 3 appear more reserved and raise a few partially overlapping concerns.

Reviewer 1 notes the lack of mechanistic understanding on how RT affects PD1 and suggests various potential approaches. S/he stresses the importance of defining the events leading to the therapeutic response.

Reviewer 3 is concerned about the lack of information on the type of infiltrating CD4+ cells (effector CD4+ and Treg), which is an important issue in this context. S/he also notes that the manuscript would benefit from streamlining to make it more readable and accessible. I especially agree with this point.

The three reviewers also list other items for your action. I would like to point out that two reviewers (and then the third as well during cross-commenting) do wonder about the usefulness and translational relevance of the mathematical model.

After reviewer cross-commenting and further discussion, it was agreed that you should be allowed to submit a revised manuscript, with the understanding that the Reviewers' concerns must be addressed with additional experimental data where appropriate and that acceptance of the manuscript will entail a second round of review. It also emerged that ultimately, the lack of the mechanistic data would not directly preclude translational development/clinical implication; I do, however encourage you to develop your study as far as realistically possible in a mechanistic sense for your next, revised version to strengthen your findings and increase their global impact. Regarding the mathematical model, I would encourage you to provide a much better explanation of its actual relevance or, in alternative, to omit the data altogether.

It is important that you consider that it is EMBO Molecular Medicine policy to allow a single round of revision only and that, therefore, acceptance or rejection of the manuscript will depend on the completeness of your responses included in the next, final version of the manuscript.

As you know, EMBO Molecular Medicine has a "scooping protection" policy, whereby similar findings that are published by others during review or revision are not a criterion for rejection. However, I do ask you to get in touch with us after three months if you have not completed your revision, to update us on the status. Please also contact us as soon as possible if similar work is published elsewhere.

Finally, please note that EMBO Molecular Medicine now requires a complete author checklist (<http://embomolmed.embopress.org/authorguide#editorial3>) to be submitted with all revised manuscripts. Provision of the author checklist is mandatory at revision stage; The checklist is designed to enhance and standardize reporting of key information in research papers and to support reanalysis and repetition of experiments by the community. The list covers key information for figure panels and captions and focuses on statistics, the reporting of reagents, animal models and human subject-derived data, as well as guidance to optimise data accessibility.

Please note that we now mandate that all corresponding authors list an ORCID digital identifier. You may do so through our web platform upon submission and the procedure takes <90 seconds to complete. We also encourage co-authors to supply an ORCID identifier, which will be linked to their name for unambiguous name identification.

I look forward to seeing a revised form of your manuscript as soon as possible.

***** Reviewer's comments *****

Referee #1 (Comments on Novelty/Model System):

The study is adequate syngeneic and GEMM models to test the in vivo efficacy of anti-PD1 therapy in combination with RT or chemotherapy.

Referee #1 (Remarks):

This is a very interesting study that may help explain in part the failure of the use of Anti-PD1 therapy in pancreatic cancer and contribute to the design of the new therapeutic approaches for this malignancy. However, the manuscript lacks critical data essential to support the major conclusions of the study and define its value for future clinical trials targeting PD1. For instance, is the effect of RT on the efficacy of PD1 blockade due to an increase of neoepitopes, changes in the immune suppressive phenotype of pancreatic tumors or both? Both potential outcomes should be evaluated. RT could affect the levels/activity of number of other cytokines that could explain the effect of PD1 blockade contributing to the tumoral immune response downregulation. Additional profiling is needed to rule out the activation of the other pathways controlling the activation of T effector cells and/or immune suppressive phenotype. Is there any effect on the expression of immune suppressive molecules IL8, IL6, CXCLs, TGFβ, etc? The clinical/translational value of the of mathematical model is not clearly defined. How this will help treatment decisions? Validation of the model should be provided in vivo and in different settings to define its clinical value. Also, the characterization of the transcriptional mechanism used by Stat1 to control the expression of PDL1 should be explored. Is this a direct target of this transcription factor? What is the role of

STAT3? This is the main STAT factor activated in pancreatic cancer. Also, are the RT and Gem used the same mechanism to upregulate PDL1? Which of these ones is related to IFN γ ? Is IL6 playing a role in this phenomenon? Further and detailed characterization of this mechanism should be evaluated, this is critical to define the mode of the action of the combination as well as design future treatment regimens using anti-PDL1 therapy.

Referee #2 (Comments on Novelty/Model System):

The authors performed the experiments very well, indeed. Different mouse models were used, yielding similar results, and combinational therapies (immunotherapy combined with other strategies) is a hot topic. Pancreatic cancer is still a tumor with a high medical need. I can, however, not judge the mathematical model, since I am not a specialist for bioinformatical modeling.

Referee #2 (Remarks):

The key finding of the current manuscript by Azad et al. is that radiotherapy renders pancreatic cancer, a normally PD-L1-blockade resistant tumor, sensitive for this kind of treatment. Due to the high therapeutic demand of this malignancy, this is of high interest to the immunotherapeutic community. The paper is very well written, the experiments are convincing, and the authors also provide functional and mechanistic data up to mathematical modeling. Hence I have only few suggestions how the authors might further improve the manuscript:

- 1) Since the field is indeed rapidly moving, almost all publications dealing with similar strategies are younger than one year. The authors should do an update on the cited literature.
- 2) On page 9, first paragraph, the last sentence starts with a typo
- 3) The effect of lower radiation doses (esp. 5 x 2gy) fail to produce significant differences, however, there seems to be an effect. (Not-significant does not mean it is the same, you just cannot prove that it is different.) Hence I would not say that effects were only present with higher doses and modify the discussion on page 11, 2nd paragraph accordingly.
- 4) If possible, please use colors for the tumor-volume kinetics, which are easier to distinguish.

I would like to finish by congratulating the authors for such an excellent manuscript.

Referee #3 (Remarks):

In this manuscript, the authors are investigating whether combining PD-L1 inhibition and radio- and/or chemo- therapies could sensitize pancreatic adenocarcinoma to treatment. This approach is particularly important as pancreatic cancers are of poor prognosis and resistant to single-agent checkpoint inhibitors or radiotherapy. Therefore, they analysed the effect of anti PD-L1 administration on tumor response to low/high, single/fractionated RT doses and the timing of single dose RT relative to anti-PD-L1 administration. The authors showed that anti-PD-L1 induced a significant growth delay after high but not low single or fractionated RT doses, an effect dependent on CD8⁺ T cells and IFN γ and that a triple combination with chemotherapy further increased this effect. They also showed that the timing of PD-L1 administration relative to RT is important for optimal effect.

Despite a solid work, the overall content in terms of quality and originality is a little bit disappointing.

The review of the literature concerning combining therapies in PDAC could have been more exhaustive.

To my point of view, to be fully exploited, this work would have deserved more extensive investigations.

For instance, infiltration of CD4⁺ T cells has been analysed but without distinguishing effector CD4⁺ and Treg. Yet, the influence of PD1/PDL1 therapy on infiltrated Treg is debated and CD8⁺ T cells/Treg ratio has been described as an important feature for treatment efficacy: It would have been interesting to distinguish CD3⁺ CD4⁺ Foxp3⁺ and Foxp3⁻.

As well, since PD1/PD-L1 axis has been shown to be involved in the reversal of T-cell exhaustion, a deeper analysis of the activation status of infiltrated T (CD69⁺, CD44⁺, CD62L⁻, Fas ligand) cells could provide additional informations. The results presented in figure 5 on peripheral T cells as well

as the discussion are not convincing.

The mathematical model is presented as a novelty to explain the results but for non familiar readers, its principles are rather obscure as the terms are not explain, even briefly and thus this paragraph is of no real interest as such.

Moreover, the surfeit of figures, both in the core paper and in the supplementary data, leads to a feeling of overload which dilutes the informations.

In conclusion, the study is interesting and addresses several relevant points such as several models (tumor and metastsases), timing of immunotherapy administration relative to RT. But the manuscript should be revised by removing redundant data that weaken the message and by highlighting the real novelties compared to the literature.

Other Comments:

- Most of the figures should be lighten : it is not necessary to present dot-plots + % gated cells in the core paper
- upregulation of PD-L1 in vivo after RT should be better in figure 1 than in supplementary figure as most of the combined effect are investigated in vivo.
- In supplementary Fig5 , legends and statistic tests are not coherent (12GY+ aPDL1 versus 5x3Gy)
- What is the rational to investigate IFNg expression in PDAC tumor cells after RT or gemcitabine treatment as it is normally secreted by immune cells ?

Concerning the in vivo experiments:

- the endpoint of the experiments is not clear: tumor burden too high? mice survival ?
- the statistic analysis used in each experiment should be mentioned in the figure legends
- may be a naive question: is the irradiation local or whole-body? does this could make any difference?
- experiment with very high RT dose could shift in supplementary data
- for sake of clarity, arrows could be added in the figures to indicate PD-L1 treatment
- in some figure, time to volume is 400 mm³, in other it is 500 mm³, it should be harmonized, as well as the y scale. Moreover, in Fig2A curves and histogrammes are different for control and aPD-L1.

Minor points:

FACS should be replaced by cytometry

p7, Additionally IFN we assessed..., IFN should be removed.

1st Revision - authors' response

04 October 2016

1st Reviewer:

The study is adequate syngeneic and GEMM models to test the in vivo efficacy of anti-PD1 therapy in combination with RT or chemotherapy.

Comment #1: This is a very interesting study that may help explain in part the failure of the use of Anti-PD1 therapy in pancreatic cancer and contribute to the design of the new therapeutic approaches for this malignancy. However, the manuscript lacks critical data essential to support the major conclusions of the study and define its value for future clinical trials targeting PD1. For instance, is the effect of RT on the efficacy of PD1 blockade due to an increase of neoepitopes, changes in the immune suppressive phenotype of pancreatic tumors or both? Both potential outcomes should be evaluated.

Reply to the Comment #1: *We would like to thank the reviewer for the important comment. We have focused on the immunosuppressive microenvironment to address at least one of the two possible mechanisms. Our main findings indicated that radiotherapy induces infiltration of immune cells including CD8⁺ T cells to the tumor microenvironment, whilst the combination of radiotherapy with anti-PD-L1 significantly reduced immune suppressive cells including CD11b⁺Gr1⁺ myeloid cells (characteristic of myeloid derived suppressor cells) and CD4⁺CD25⁺FoxP3⁺ T regulatory cells in the allograft and liver metastasis models.*

We have now repeated the triple combination in vivo experiment using Gemcitabine+radiotherapy+anti-PD-L1 in the KPC allografts and analysed T cell activation markers (as requested by the 3rd reviewer). In line with our previous data in peripheral blood (now removed per Reviewer's request), addition of anti-PD-L1 to radiotherapy resulted in increased expression of the activation markers CD69, CD44 and FasL on tumor-infiltrating CD8⁺ T cells (Figure 5B). Moreover, addition of anti-PD-L1 to RT led to increased CD8/Tregs ratio in the triple

combination experiment (Figure 5C). Furthermore, we repeated the liver metastasis experiment and confirmed decrease in the CD4+CD25+FoxP3+ Tregs infiltration following treatment with RT+anti-PD-L1 (Figure 6B and Supplementary Figure 10). Altogether, these data indicate that radiation in combination with PD-L1 checkpoint inhibition can reverse the immune suppressive tumor environment and “shift” the balance towards a more favourable immune phenotype. The Materials & Methods (Page 23, second and third paragraph), Results (Page 10, second paragraph; page 11, first paragraph), Discussion (Page 13, second paragraph; page 15, last paragraph; page 17, second paragraph), Main and Supplementary Figure legends have been modified accordingly. Although we agree with the reviewer that neoepitope analysis is an important factor in the efficacy of PD-L1 blockade, this analysis is beyond the scope of the study as it would necessitate extensive exome and transcriptome sequencing of the tumors, and together with the analysis and validation of possible sequencing targets, would require at least 6 months of work; this is very difficult to achieve given the journal's deadline for resubmission of only 3 months.

Comment #2: RT could affect the levels/activity of number of other cytokines that could explain the effect of PD1 blockade contributing to the tumoral immune response downregulation. Additional profiling is needed to rule out the activation of the other pathways controlling the activation of T effector cells and/or immune suppressive phenotype. Is there any effect on the expression of immune suppressive molecules IL8, IL6, CXCLs, TGFβ, etc?

Reply to the Comment #2: We have performed cytokine profiling of serum from tumor-bearing mice treated as shown in Supplementary Figure 8. We found decreased expression of stromal derived factor 1 (SDF-1) and IL-1RA after radiotherapy+anti-PD-L1 compared to controls but no differences in expression of other immune suppressive molecules (Supplementary Figure 8A-B). In particular, ELISA assay showed that SDF-1 was significantly decreased after radiotherapy, and was further reduced when combined with anti-PD-L1 (Supplementary Figure 8C). SDF-1 has been previously implicated in immunosuppression in PDAC and its downregulation in our model may contribute to reversing the immune suppressive phenotype (Feig et al.). The Materials & Methods (Page 24, second paragraph), Results (Page 9, first paragraph), Discussion (Page 16, last paragraph) and Supplementary Figure legend have been modified accordingly.

Comment #3: The clinical/translational value of the mathematical model is not clearly defined. How this will help treatment decisions? Validation of the model should be provided in vivo and in different settings to define its clinical value.

Reply to the Comment #3: We developed this minimal model of immune cell interactions with tumor cells to explain tumor growth behaviour under treatment conditions and to provide additional insight into the underlying mechanisms of synergy between radiotherapy and anti-PD-L1 (Supplementary Figure 11A-G). Specifically, the current mathematical model is able to explain the key observations that PD-L1 blockade leads to efficient killing of irradiated (susceptible) tumor cells, an effect which is lost upon CD8 T cell depletion or if anti-PD-L1 is administered 7 days after RT. This has valuable clinical implications in helping to guide treatment decisions; our model indicates that radiation+anti-PD-L1 combination therapy is only effective in patients with pre-existing CD8 T cells in the tumour microenvironment, and that anti-PD-L1 should be administered early, in conjunction with radiotherapy for maximal anti-tumor responses.

Our mathematical model also highlighted that the synergistic effects of anti-PD-L1 treatment with radiotherapy cannot be explained solely with increased CD8 T cell recruitment or activation. An additional effect of anti-PD-L1 on the immunosuppressive microenvironment had to be invoked to explain the delayed tumor growth. Hence, the mathematical model provides validation that anti-PD-L1 additionally targets the immunosuppressive cells whereas radiotherapy alone may only stimulate immune cell recruitment. These findings are important for the design of future therapeutic strategies as they underscore the clinical value of combination therapy involving PD-1/PD-L1 blockade to target the immunosuppressive microenvironment. We have now modified our Results (Page 11, last paragraph; Page 12-page 13, first paragraph) and Discussion (Page 19, first paragraph) sections to underline the significance of the mathematical model. The mathematical model data are now shown as Supplementary Figure 11 with the corresponding Figure legend.

Comment #4: Also, the characterization of the transcriptional mechanism used by Stat1 to control the expression of PDL1 should be explored. Is this a direct target of this transcription factor? What is the role of STAT3? This is the main STAT factor activated in pancreatic cancer.

Reply to the Comment #4: To address this comment we have downregulated Stat1 and Stat3 expression using siRNA in our PDAC cells (Supplementary Figure 1F-G). Inhibition of Stat1 but not Stat3 decreased PD-L1 expression after radiotherapy and gemcitabine treatment in PDAC cells

(Figure 1E-F; Supplementary Figure 1H). Our data suggests that Stat1 but not Stat3 signalling regulates PD-L1 expression, and that Stat1 is activated upon radiotherapy and gemcitabine treatment in PDAC cells to induced PD-L1 expression. The Methods (Page 22, last paragraph; page 23, first paragraph), Results (Page 5, first paragraph), Discussion (Page 15, second paragraph), Main and Supplementary Figure legend sections have been modified to include the new findings.

Comment #5: Also, are the RT and Gem used the same mechanism to upregulate PDL1? Which of these ones is related to IFNgamma?

Reply to the Comment #5: In our study, we found that both radiotherapy and gemcitabine treatment induced activation of the Jak/Stat pathway in PDAC cells (Figure 1C-D), which coincided with PD-L1 upregulation. Inhibition of Jak/Stat signalling using AG490 completely abrogated PD-L1 upregulation. Because AG490 inhibits both Stat1 and Stat3, we performed additional experiments (using siRNA) to show that Stat1 but not Stat3 was responsible for PD-L1 upregulation.

Several inflammatory mediators including IFNgamma can activate the Jak/Stat1 pathway. We did not detect changes in IFNgamma expression after either siRNA for Stat1 or Stat3 (Supplementary Figure 3B-C). Although radiotherapy and gemcitabine treatment stimulated Jak/Stat signalling, only gemcitabine treatment (and not radiotherapy) upregulated IFNgamma expression in PDAC cells (Supplementary Figure 3A). Our data suggest that gemcitabine treatment may upregulate IFNgamma to activate the Jak/Stat pathway, thus inducing PD-L1 expression whereas radiotherapy-mediated activation of the Jak/Stat pathway may occur via other inflammatory mediators. We have included the new data in the Page 22, last paragraph; page 23, first paragraph), Results (Page 5, first and second paragraphs), Discussion (Page 15, second paragraph; page 16, last paragraph) and Supplementary Figure legend sections.

Comment #6: Is IL6 playing a role in this phenomenon? Further and detailed characterization of this mechanism should be evaluated, this is critical to define the mode of the action of the combination as well as design future treatment regimens using anti-PDL1 therapy.

Reply to the Comment #6: To address this issue, we analysed PD-L1 expression on untreated, irradiated and gemcitabine-treated PDAC cells either in the presence or absence of recombinant IL-6. We found that addition of IL-6 did not alter PD-L1 (Supplementary Figure 3A) or IFN γ expression (Supplementary Figure 3B) regardless of treatment. Moreover, we did not detect any significant changes in IL-6 levels in sera of KPC tumor-bearing mice following treatment (Supplementary Figure 8A-B). Our data suggests that IL-6 is unlikely to play a major role in this phenomenon. We have included the new data in the Methods (Page 23, first paragraph; Page 24, second paragraph), Results (Page 5, last paragraph; Page 9, first paragraph), Discussion (Page 15, second paragraph; Page 16, last paragraph) and Supplementary Figure legend sections.

2nd Reviewer:

The authors performed the experiments very well, indeed. Different mouse models were used, yielding similar results, and combinational therapies (immunotherapy combined with other strategies) is a hot topic. Pancreatic cancer is still a tumor with a high medical need. I can, however, not judge the mathematical model, since I am not a specialist for bioinformatical modeling. The key finding of the current manuscript by Azad et al. is that radiotherapy renders pancreatic cancer, a normally PD-L1-blockade resistant tumor, sensitive for this kind of treatment. Due to the high therapeutic demand of this malignancy, this is of high interest to the immunotherapeutic community. The paper is very well written, the experiments are convincing, and the authors also provide functional and mechanistic data up to mathematical modeling. Hence I have only few suggestions how the authors might further improve the manuscript:

Comment #1: Since the field is indeed rapidly moving, almost all publications dealing with similar strategies are younger than one year. The authors should do an update on the cited literature.

Reply to the Comment #1: Many thanks for the positive feedback. We have now edited the publication list to include the latest articles. New references (Rodriguez-Ruiz et al. 2016; Beatty et al 2015; Diana et al 2016; Zhang et al 2016; Amedei et al 2013; Lutz et al 2014; Sharabi et al 2015a+b; Feig et al. 2013; Kim et al 2016; Belluci et al 2015; Burnette et al 2013; Byrne et al 2011; Chen et al 2016), have been inserted and discussed in the manuscript.

Comment #2: On page 9, first paragraph, the last sentence starts with a typo

Reply to the Comment #2: This typo has been corrected.

Comment #3: The effect of lower radiation doses (esp. 5 x 2gy) fail to produce significant differences, however, there seems to be an effect. (Not-significant does not mean it is the same, you just cannot prove that it is different.) Hence, I would not say that effects were only present with higher doses and modify the discussion on page 11, 2nd paragraph accordingly.

Reply to the Comment #3: *We agree and have modified the Results (Page 6, last paragraph) and Discussion (Page 13, last paragraph) and Figure legend as recommended.*

Comment #4: If possible, please use colors for the tumor-volume kinetics, which are easier to distinguish. I would like to finish by congratulating the authors for such an excellent manuscript.

Reply to the Comment #4: *Thank you, tumor volume growth curves are now shown in color.*

3rd Reviewer:

In this manuscript, the authors are investigating whether combining PD-L1 inhibition and radio- and/or chemo- therapies could sensitize pancreatic adenocarcinoma to treatment. This approach is particularly important as pancreatic cancers are of poor prognosis and resistant to single-agent checkpoint inhibitors or radiotherapy. Therefore, they analysed the effect of anti PD-L1 administration on tumor response to low/high, single/fractionated RT doses and the timing of single dose RT relative to anti-PD-L1 administration. The authors showed that anti-PD-L1 induced a significant growth delay after high but not low single or fractionated RT doses, an effect dependent on CD8+ T cells and IFN γ and that a triple combination with chemotherapy further increased this effect. They also showed that the timing of PD-L1 administration relative to RT is important for optimal effect. Despite a solid work, the overall content in terms of quality and originality is a little bit disappointing.

Comment #1: The review of the literature concerning combining therapies in PDAC could have been more exhaustive.

Reply to the Comment #1: *We have now edited the Introduction and Discussion sections to include a more comprehensive and updated review on combination therapies in PDAC (Rodriguez-Ruiz et al. 2016; Beatty et al 2015; Diana et al 2016; Zhang et al 2016; Amedei et al 2013; Lutz et al 2014; Sharabi et al 2015a+b; Feig et al. 2013; Kim et al 2016; Belluci et al 2015; Burnette et al 2013; Byrne et al 2011; Chen et al 2016).*

Comment #2: To my point of view, to be fully exploited, this work would have deserved more extensive investigations. For instance, infiltration of CD4+ T cells has been analysed but without distinguishing effector CD4+ and Treg. Yet, the influence of PD1/PDL1 therapy on infiltrated Treg is debated and CD8+ T cells/Treg ratio has been described as an important feature for treatment efficacy: It would have been interesting to distinguish CD3+ CD4+ Foxp3+ and Foxp3- .

Reply to the Comment #2: *We would like to thank the reviewer for the important comment. We have now included a more extensive investigation on the infiltrating T cell populations in the KPC allograft (with triple combination treatment) and liver metastasis mouse models. Specifically, we have looked at activated effector CD8+ T cells (expressing CD69, CD44, and FasL), the CD8+ T cells/Tregs ratio and the CD4+CD25+FOXP3+ T regulatory (Tregs) cells in these new sets of experiments (Figures 5B-C and 6B; Supplementary Figure Supplementary Figure 10).*

We found that radiotherapy induces infiltration of immune cells including CD8+ T cells to the tumor microenvironment, whilst the combination of radiotherapy with anti-PD-L1 significantly reduced immune suppressive cells including CD11b+Gr1+ myeloid cells (characteristic of myeloid derived suppressor cells) in the allograft model (Figure 3). In the KPC allograft model (Figure 5), addition of anti-PD-L1 to radiotherapy resulted in increased expression of activation markers CD69, CD44 and FasL on tumor-infiltrating CD8+ T cells (Figure 5B) and increased CD8/Treg ratio (Treg defined as CD4+CD25+FOXP3+ cells; Figure 5C). Furthermore, in the liver metastasis experiment, RT+anti-PD-L1 resulted in decreased infiltration by CD4+CD25+FOXP3+ Tregs.

Overall, our data indicates that radiation in combination with PD-L1 checkpoint inhibition can reverse the immune suppressive tumor environment and “shift” the balance towards a more favourable immune phenotype. The Materials & Methods (Page 23, second and third paragraph), Results (Page 10, second paragraph; page 11, first paragraph), Discussion (Page 13, second paragraph; page 15, last paragraph; page 17, second paragraph), Main and Supplementary Figure legends have been modified accordingly.

Comment #3: As well, since PD1/PD-L1 axis has been shown to be involved in the reversal of T-cell exhaustion, a deeper analysis of the activation status of infiltrated T (CD69+, CD44+, CD62L-, Fas ligand) cells could provide additional information. The results presented in figure 5 on peripheral T cells as well as the discussion are not convincing.

Reply to the Comment #3: *We agree with the reviewer and have now provided a more comprehensive characterisation of T cell activation status. Specifically, we have examined expression of the T cell activation markers CD69, CD44, and Fas ligand on the infiltrating CD8+ T cell population (Figure 5B), as replied above (Reply to the 2nd comment).*

Comment #4: The mathematical model is presented as a novelty to explain the results but for non-familiar readers, its principles are rather obscure as the terms are not explain, even briefly and thus this paragraph is of no real interest as such.

Reply to the Comment #4: *We believe that the mathematical model addresses the underlying mechanisms of synergy between radiotherapy and anti-PD-L1 and provides clinical insights that may help guide treatment decisions (Supplementary Figure 11A-G). Our model explains the key observations that PD-L1 blockade leads to efficient killing of irradiated (susceptible) tumor cells, an effect which is lost upon CD8 T cell depletion or if anti-PD-L1 is administered 7 days after radiotherapy. This has valuable clinical implications in helping to guide treatment decisions; our model indicates that radiation+anti-PD-L1 combination therapy is only effective in patients with pre-existing CD8 T cells in the tumour microenvironment, and that anti-PD-L1 should be administered early, in conjunction with radiotherapy for maximal anti-tumor responses.*

Our mathematical model also highlighted that the synergistic effects of anti-PD-L1 treatment with radiotherapy cannot be explained solely with increased CD8 T cell recruitment or activation. An additional effect of anti-PD-L1 on the immunosuppressive microenvironment had to be invoked to explain the delayed tumor growth. Hence, the mathematical model provides validation that anti-PD-L1 additionally targets the immunosuppressive cells whereas radiotherapy alone may only stimulate immune cell recruitment. These findings are important for the design of future therapeutic strategies as they underscore the clinical value of combination therapy involving PD-1/PD-L1 blockade to target the immunosuppressive microenvironment.

We have now modified our Results (Page 11, last paragraph; Page 12-page 13, first paragraph) and Discussion (Page 19, first paragraph) sections to underline the significance of the mathematical model. The mathematical model data are now shown as Supplementary Figure 11 with the corresponding Figure legend.

Comment #5: Moreover, the surfeit of figures, both in the core paper and in the supplementary data, leads to a feeling of overload which dilutes the information. In conclusion, the study is interesting and addresses several relevant points such as several models (tumor and metastases), timing of immunotherapy administration relative to RT. But the manuscript should be revised by removing redundant data that weaken the message and by highlighting the real novelties compared to the literature.

Reply to the Comment #5: *We thank the reviewer for the insightful comment and have now removed redundant data and figures (e.g. Figure 5C, Supplementary Figure 1B, Supplementary Figure 2, Supplementary Figure 3 and Supplementary Figure 4 in the first submission). In addition, flow cytometry dots plots and histograms have now been moved to Supplementary Figures 5, 6 and 10. We have also streamlined and re-written the paper considerably to avoid “overloading” and to further emphasize the significance of our findings.*

Comment #6: Most of the figures should be lighten: it is not necessary to present dot-plots + % gated cells in the core paper.

Reply to the Comment #6: *We have moved the dot plots and histograms from our flow cytometry analysis to Supplementary Figures to lighten the paper as suggested (Supplementary Figures 5, 6 and 10).*

Comment #7: Upregulation of PD-L1 in vivo after RT should be better in figure 1 than in supplementary figure as most of the combined effects are investigated in vivo.

Reply to the Comment #7: *We intended to include results of our in vitro experiments in Figure 1 and present our results of PD-L1 upregulation in vivo in subsequent figure (Supplementary Figure 9B-C) so as to distinguish between the in vitro and in vivo studies. Moreover, the in vivo studies are described later in the text and as such, we prefer to keep this result in Supplementary Figure 9 to avoid possible confusion. We appreciate your consideration and hope you agree with this decision.*

Comment #8: In supplementary Fig5, legends and statistic tests are not coherent (12GY+ aPDL1 versus 5x3Gy)

Reply to the Comment #8: *We apologize for the oversight and have now corrected the figure accordingly (Supplementary Figure 7A-D).*

Comment #9: What is the rationale to investigate IFN γ expression in PDAC tumor cells after RT or gemcitabine treatment as it is normally secreted by immune cells?

Reply to the Comment #9: *We agree with the reviewer that IFN γ is predominantly produced by immune cells in the tumor microenvironment. In our study, we show that radiotherapy and gemcitabine treatment stimulated PD-L1 expression in a JAK/Stat1-dependent manner. Previous studies have shown that IFN γ can induce Stat1 activation in tumor cells. Following on from this, we assessed whether radiotherapy and gemcitabine can upregulate IFN γ expression in tumor cells to induce Stat1 signalling (Supplementary Figure 2). We have included the new data in the Page 22, last paragraph; page 23, first paragraph), Results (Page 5, first and second paragraphs), Discussion (Page 15, second paragraph; page 16, last paragraph) and Supplementary Figure legend sections.*

Comment #10: Concerning the in vivo experiments: - the endpoint of the experiments is not clear: tumor burden too high? mice survival?

Reply to the Comment #10: *We used endpoints that will give us readable data whilst ensuring that our experiments are according to the UK Home Office guidelines for animal welfare. Based on these guidelines and our animal licence, the maximum tumor volume allowed was 750mm³. Treatments were initiated once subcutaneous tumors reached a size of 100mm³ and mice were sacrificed (endpoint) once the tumor reached a volume of approximately 500-600mm³. We have now re-analysed the time it took for tumors to reach a volume of 400 mm³ i.e. 4 times the starting treatment volume so as to present a standardized calculation across all in vivo experiments (See reply to comment 15). We have inserted this information in the revised manuscript to avoid any confusion (Page 21, first paragraph).*

Comment #11: the statistic analysis used in each experiment should be mentioned in the figure legends.

Reply to the Comment #11: *Details on the statistical analysis used have now been included in the Main and Supplementary Figure legends.*

Comment #12: may be a naive question: is the irradiation local or whole-body? does this could make any difference?

Reply to the Comment #12: *Radiotherapy is given locally, targeting the subcutaneous tumors as described in a previous publication (Fokas et al. Cancer Research 2012). The rest of the body was sealed to avoid unwanted side effects. We have included this detail in the revised manuscript (Page 21, first paragraph) to avoid any misunderstanding.*

Comment #13: experiment with very high RT dose could shift in supplementary data.

Reply to the Comment #13: *We have now moved the results from this experiment to Supplementary Data (Supplementary Figure 9A) and have modified our Results section and Figure legend accordingly.*

Comment #14: for sake of clarity, arrows could be added in the figures to indicate PD-L1 treatment.

Reply to the Comment #14: *This has now been added to Figures 2-5.*

Comment #15: in some figure, time to volume is 400 mm³, in other it is 500 mm³, it should be harmonized, as well as the y scale. Moreover, in Fig2A curves and histograms are different for control and aPD-L1.

Reply to the Comment #15: *We have now changed Figure 5A to standardize our calculation of tumor growth as time to 400mm³ in all in vivo experiments (Figures 2, 4 and 5).*

Comment #16: FACS should be replaced by cytometry

Reply to the Comment #16: *This point has been corrected accordingly.*

Comment #17: p7, Additionally IFN we assessed..., IFN should be removed.

Reply to the Comment #17: *This error has been corrected accordingly.*

2nd Editorial Decision

19 October 2016

Thank you for the submission of your revised manuscript to EMBO Molecular Medicine. We have now received the enclosed reports from the referees that were asked to re-assess it. As you will see the reviewers are now globally supportive and I am pleased to inform you that we will be able to accept your manuscript pending the following final amendments:

- 1) Please carefully address reviewer 3's final concerns. I am prepared to deal with your revision at the editorial level provided you take action on each point by amending the manuscript or providing an explanation
- 2) As per our Author Guidelines, the description of all reported data that includes statistical testing must state the name of the statistical test used to generate error bars and P values, the number (n) of independent experiments underlying each data point (not replicate measures of one sample), and the actual P value for each test (not merely 'significant' or 'P < 0.05').
- 3) We note that all figures are presented in landscape orientation. Please reorganize into portrait mode.
- 4) Please rename Supplementary figures to Appendix Figure S1, S2, etc. and combine legends and images to create a single PDF file. If instead you chose to present them as EV figures (see guide to authors: <http://embomolmed.embopress.org/authorguide#expandedview>) then orientation must be changed to portrait mode.
- 5) The manuscript must include a statement in the Materials and Methods identifying the institutional and/or licensing committee approving the experiments, including any relevant details (like how many animals were used, of which gender, at what age, which strains, if genetically modified, on which background, housing details, etc). We encourage authors to follow the ARRIVE guidelines for reporting studies involving animals. Please see the EQUATOR website for details: <http://www.equator-network.org/reporting-guidelines/improving-bioscience-research-reporting-the-arrive-guidelines-for-reporting-animal-research/>. Please make sure that ALL the above details are reported.
- 6) We encourage the publication of source data, particularly for electrophoretic gels and blots, with the aim of making primary data more accessible and transparent to the reader. Would you be willing to provide a PDF file per figure that contains the original, uncropped and unprocessed scans of all or at least the key gels used in the manuscript? The PDF files should be labeled with the appropriate figure/panel number, and should have molecular weight markers; further annotation may be useful but is not essential. The PDF files will be published online with the article as supplementary "Source Data" files. If you have any questions regarding this just contact me.
- 7) Every published paper now includes a 'Synopsis' to further enhance discoverability. Synopses are displayed on the journal webpage and are freely accessible to all readers. They include a short standfirst as well as 2-5 one sentence bullet points that summarise the paper. Please provide the

synopsis including the short list of bullet points that summarise the key NEW findings. The bullet points should be designed to be complementary to the abstract - i.e. not repeat the same text. We encourage inclusion of key acronyms and quantitative information. Please use the passive voice. Please attach this information in a separate file or send them by email, we will incorporate it accordingly. You are also welcome to suggest a striking image or visual abstract to illustrate your article. If you do please provide a jpeg file 550 px-wide x 400-px high.

Please submit your revised manuscript within two weeks. I look forward to seeing a revised form of your manuscript as soon as possible.

***** Reviewer's comments *****

Referee #1 (Comments on Novelty/Model System):

The manuscript has sufficiently improved since the initial submission. However, it is a pity that some aspects are still (in my view) under developed. Specially the mathematical model which could have tremendous applications in the clinical setting. Understanding tumor heterogeneity and evolution using a model of this kind will help tremendously the current precision medicine efforts. Nonetheless, the manuscript is excellent and the authors have been responsive to reviewers critiques. The study will be of high interest for the readership of EMM. I recommend to make some sort of highlight of the article. Finally, there are minor suggestions that I have regarding the text that I will leave up to the editor to decide if the authors need to address them. These are minor writing changes that may help define the significance and impact of the study, and their understanding of the findings.

Referee #1 (Remarks):

As stated in my previous review, this is an excellent study and highly significance for the field of precision medicine. The authors have been responsive to the reviewers critiques and the conclusions of the study are strongly supported the new data included in the revised version of the manuscript. There are minor changes in the writing that may help define the impact of the studies for precision medicine and its usefulness for individualized treatments. For example, the mathematical model as described may be difficult to fully appreciate by the non-expert (e.g., practicing oncologist)

Referee #3 (Remarks):

The authors have taken into account most of my comments and have successfully updated the bibliography and lightened both text and figures. Moreover, data on infiltrating T cell populations, in particular concerning the ratio CD8+/Treg and the activation status have been added, reinforcing the relevant points addressed by this work.

Concerning the mathematical model, I am still not fully convinced but efforts have been made to better explain its interest, modifying the text accordingly. In this respect, I would like to mention that the page umbering used by the authors for the inserted modifications is not in agreement with that of the version I got, that does not facilitate the task.

Minor points to be corrected:

- In figure 2A, histogramme and curves do not fit. This has not been corrected : Control (400mm³): 4 days on curve/6 in histogramme; 6 days for aPD-L1 versus >10..
- In supplementary figure 1F, the concentration of stat3 siRNA is not mentioned, nor in the legend.
- In 1G, Stat3-siRNA-50nM should be corrected for Stat1-si RNA
- p6: "resulting" in large wounds should be "resulted in"
- supplementary figure 8: what is the y axis legend?
- Supplementary Figure 9B and C: 1 star between 12 gy and 5x3gy and no star between 5x3 and 20 gy ?? 3 stars between control/gemcitabine short term? Do you confirm the statistic?
- p9: significantly increased numbers of ; after 14 days, livers were harvested and weighted
- p9, end of first paragraph: These data provide evidence on the potential of PD-L1 blockade to promote T cell activation in combination with RT and gemcitabine.

From Figure 5: PDL1 blockade alone significantly reverses CD8/Treg ratio compared to control, RT or gemcitabine treatment . Combined treatment decreases the ratio compared to anti-PDL1 alone,

and addition of anti-PDL1 to combined treatment increases the CD8/Treg ratio.
So it will more accurate to say that '...;PD-L1 blockade to promote T cell activation in RT, gemcitabine or combined treatment'

- p14: appears

- p17: kg instead of Kg

2nd Revision - authors' response

26 October 2016

1st Reviewer:

Comment #1: The manuscript has sufficiently improved since the initial submission. However, it is a pity that some aspects are still (in my view) under developed. Specially the mathematical model which could have tremendous applications in the clinical setting. Understanding tumor heterogeneity and evolution using a model of this kind will help tremendously the current precision medicine efforts. Nonetheless, the manuscript is excellent and the authors have been responsive to reviewers critiques. The study will be of high interest for the readership of EMM. I recommend to make some sort of highlight of the article. Finally, there are minor suggestions that I have regarding the text that I will leave up to the editor to decide if the authors need to address them. These are minor writing changes that may help define the significance and impact of the study, and their understanding of the findings.

Referee #1 (Remarks): As stated in my previous review, this is an excellent study and highly significance for the field of precision medicine. The authors have been responsive to the reviewers critiques and the conclusions of the study are strongly supported the new data included in the revised version of the manuscript. There are minor changes in the writing that may help define the impact of the studies for precision medicine and its usefulness for individualized treatments. For example, the mathematical model as described may be difficult to fully appreciate by the non-expert (e.g., practicing oncologist)

Reply to the Comment #1: *We would like to thank the reviewer for the positive feedback. We have re-written the mathematical model results (Page 10, second paragraph to page 12, first paragraph) to further enhance understanding and appreciation of its findings, and its implications in era of precision medicine. The panel order of Appendix Figure S11 and its legend have been modified accordingly.*

3rd Reviewer:

General comment: The authors have taken into account most of my comments and have successfully updated the bibliography and lightened both text and figures. Moreover, data on infiltrating T cell populations, in particular concerning the ratio CD8+/Treg and the activation status have been added, reinforcing the relevant points addressed by this work. Concerning the mathematical model, I am still not fully convinced but efforts have been made to better explain its interest, modifying the text accordingly. In this respect, I would like to mention that the page umbering used by the authors for the inserted modifications is not in agreement with that of the version I got, that does not facilitate the task.

Reply to general comment: We would like to thank the reviewer for the positive feedback. We have re-written the mathematical model results (Page 10, second paragraph to page 12, first paragraph) to further enhance understanding and appreciation of its findings, and its implications in era of precision medicine. The panel order of Appendix Figure S11 and its legend have been modified accordingly.

Minor points to be corrected:

Comment #1: - In figure 2A, histogramme and curves do not fit. This has not been corrected:
Control (400mm³): 4 days on curve/6 in histogramme; 6 days for aPD-L1 versus >10.

Reply to the Comment #1: *We thank you for the careful notice and apologize for the inconvenience as we initially calculated the time to 500mm³. We have now corrected the calculation as requested. As before, PD-L1 blockade did not affect tumor growth, whereas RT radiosensitized tumors*

compared to untreated controls. Importantly, in line to our initial findings, addition of antiPD-L1 to either RT schedule did not radiosensitize tumors. Hence, our main finding remains unaltered. The figure and Results (Page 5, last paragraph) have been modified accordingly.

Comment #2: - In supplementary figure 1F, the concentration of stat3 siRNA is not mentioned, nor in the legend.

Reply to the Comment #2: *The concentration of Stat3 siRNA was 50 nM. We have included the concentration into Supplementary Figure 1F and its legend as requested. We have also inserted the concentration on Supplementary Figure 2 legend.*

Comment #3: - In 1G, Stat3-siRNA-50nM should be corrected for Stat1-si RNA

Reply to the Comment #3: *Many thanks, this error has now been corrected.*

Comment #4 - p6: "resulting" in large wounds should be "resulted in"

Reply to the Comment #4: *Thank you, this error has now been corrected,*

Comment #5: - supplementary figure 8: what is the y axis legend?

Reply to the Comment #5: *Many thanks for the careful notice. This illustrates relative expression to the positive control on the cytokine array. We have modified the figure accordingly to include this information.*

Comment #6: Supplementary Figure 9B and C: 1 star between 12 gy and 5x3gy and no star between 5x3 and 20 gy?? 3 stars between control/gemcitabine short term? Do you confirm the statistic?

Reply to the Comment #6: *Many thanks for the notice. We have re-run all statistical analyses and now inserted the actual p-values per journal's guide to the authors. Comparison between 12 Gy and 5x3 Gy showed a $p > 0.999$ i.e. not significant. Also, comparison of control vs gemcitabine short-term showed a $p = 0.031$.*

Comment #7: - p9: significantly increased numbers of; after 14 days, livers were harvested and weighted

Reply to the Comment #7: *Many thanks. These errors have now been corrected.*

Comment #8: - p9, end of first paragraph: These data provide evidence on the potential of PD-L1 blockade to promote T cell activation in combination with RT and gemcitabine. From Figure 5: PDL1 blockade alone significantly reverses CD8/Treg ratio compared to control, RT or gemcitabine treatment. Combined treatment decreases the ratio compared to anti-PDL1 alone, and addition of anti-PDL1 to combined treatment increases the CD8/Treg ratio. So it will more accurate to say that '...; PD-L1 blockade to promote T cell activation in RT, gemcitabine or combined treatment'

Reply to the Comment #8: *Many thanks. We agree with the comment and have modified the text accordingly (Page 9, second paragraph).*

Comment #9: - p14: appears

Reply to the Comment #9: *Thank you, this error has now been corrected accordingly.*

Comment #10: - p17: kg instead of Kg

Reply to the Comment #10: *Thank you, this error has now been corrected accordingly.*

Corresponding Author Name: EMMANOUIL FOKAS

Journal Submitted to: EMBO MOLECULAR MEDICINE

Manuscript Number: EMM-2016-06674